# Direct Kinetostatic Analysis of a Gripper with Curved Flexures

**DOI:** 10.3390/mi13122172

**Published:** 2022-12-08

**Authors:** Alessandro Cammarata, Pietro Davide Maddio, Rosario Sinatra, Nicola Pio Belfiore

**Affiliations:** 1Department of Civil Engineering and Architecture, University of Catania, Viale Andrea Doria, 6, 95123 Catania, Italy; 2Department of Industrial, Electronic and Mechanical Engineering, University of Roma Tre, Via Vito Volterra, 62, 00146 Rome, Italy

**Keywords:** micro-grippers, micro-manipulators, flexure, CFSH, compliant mechanism, the tangent stiffness matrix

## Abstract

Micro-electro-mechanical-systems (MEMS) extensively employed planar mechanisms with elastic curved beams. However, using a curved circular beam as a flexure hinge, in most cases, needs a more sophisticated kinetostatic model than the conventional planar flexures. An elastic curved beam generally allows its outer sections to experience full plane mobility with three degrees of freedom, making complex non-linear models necessary to predict their behavior. This paper describes the direct kinetostatic analysis of a planar gripper with an elastic curved beam is described and then solved by calculating the tangent stiffness matrix in closed form. Two simplified models and different contributions to derive their tangent stiffness matrices are considered. Then, the Newton–Raphson iterative method solves the non-linear direct kinetostatic problem. The technique, which appears particularly useful for real-time applications, is finally applied to a case study consisting of a four-bar linkage gripper with elastic curved beam joints that can be used in real-time grasping operations at the microscale.

## 1. Introduction

Compliant mechanisms have been adopted in several applications for centuries, because of several well-known advantages, such as the absence of sliding friction, backlash, and significant wear, with the advantage of requiring minimum effort assembly. The first complete and systematic description of their characteristics appeared in the literature about thirty years ago, around 1994, when Midha and Howell introduced a classification [1] that has been used until today. Another significant step forward in developing the compliant mechanisms was taken a couple of years later, when the Pseudo-Rigid-Body equivalent Model (PRBM) was introduced to evaluate the elasticity of compliant mechanisms with significant deflection capabilities [2]. More materials concerning the compliant mechanisms were presented in 2001 [3].

Compliant mechanisms are nowadays used in many fields [4], as, for instance, for the development of aerospace [5] and biomedical [6,7] devices, compliant bistable mechanisms [8,9,10,11], grasping and releasing micro-objects devices [12], precision engineering [13], MEMS [14,15,16,17,18,19], polishing and deburring [20,21,22], lab-on-chip micro and nanosystems [23], and automotive devices [24,25].

The actual configuration of a compliant mechanism depends not only on the applied forces and torques but also on its geometric characteristics, with kinematic and mechanical coupling and non-linearity problems that especially arise in case of large deformations.

A comprehensive survey on many recent techniques for modeling the kinetostatic and dynamic behavior of flexure-based compliant mechanisms has been recently presented [26].

Kinetostatic models of complex plane compliant mechanisms have been developed in both micro and macro scale devices by using a wide variety of different linear and non-linear methods, such as, for only representative example, those based on: Loop-closure equations and the static equilibrium conditions for multi-loops compliant mechanisms [27]; chained beam constraint model and geometric parameter optimization, specially conceived for translational motion [28]; a combination of a beam constraint model, load equilibrium conditions, and geometric compatibility equations, specially conceived for 3-PPR compliant parallel mechanisms [29]; a kinetostatic modeling approach that integrates the screw theory with the energy method, with consequent avoidance of the problem of finding a solution to the equilibrium equations of nodal forces and the possibility of taking into account the parasitic deformations in space [30]; an extension of the chained beam constraint model specially revisited to analyze flexible beams of effective variable length [31]; a mathematical formulation of the compliance matrix method, combined with the inverse kinematic, specially introduced for modeling the flexure-based parallel compliant mechanisms with multiple actuation forces [32]; the adoption of three representations of multiple segments 2D beam models, namely, beam constraint model, linear Euler–Bernoulli beam, and PRBM [33]; the adoption of a new two-colored digraph representation of planar flexure-based compliant mechanisms for the automatic generation of the kinetostatic equations [34]; the introduction of virtual flexure hinges, link-flexure incidence matrices, and path matrices to generate automatically the formulation of the kinetostatic equations [35].

A more specific contribution has been dedicated in 2013 [36] to the solution of the problem of inverse kinetostatic analysis of a compliant four-bar linkage with flexible circular joints and pseudo-rigid bodies. This problem was attacked by extensively applying the theory of curved beams to the flexible parts, which gave rise to the closed-form symbolic expression of the compliance matrix, and by applying the static balance equations to both the elastic and pseudo-rigid parts.

The present investigation explores the opposite problem of direct kinetostatic analysis of a planar gripper with circular flexures. Two possible models based on the static balance of flexures are provided. The first linear model considers the static equilibrium in the undeformed configuration, while the second considers the balance in the deformed configuration. Both models simplify the fully non-linear model by exploiting a constant stiffness matrix of the undeformed curved beam element. These models allow us to find the tangent stiffness matrix in closed form as the sum of different contributions. Furthermore, dividing the tangent stiffness matrix into its contributions allows for evaluating each term’s importance and setting strategies to speed up convergence. Furthermore, through a validation process of the fully non-linear model results performed on a case study, it will be possible to ascertain how the simplifications still provide accurate values in almost the entire mechanism’s range of motion. As known, the tangent stiffness matrix is the heart of an iterative solving method. It is the basis of many implicit integrators widely used for the study of flexible mechanisms, such as the generalized α-method [37] or the HHT-method [38].

The main target of this article is to create two simplified models:Solving the problem of the direct kinetostatic analysis of planar grippers with curved beams;Being reliable in terms of motion accuracy and actuation forces;Being computationally efficient to extend the formulation for real-time applications.

Any Finite Element Analysis (FEA) or Multibody Dynamics Simulation (MBDS) package is very reliable for solving any general problem in kinetostatic analysis numerically. Despite this, the availability of a ready-to-use independent algorithm to solve the direct kinetostatic problem gives rise to the possibility of implementing it in any real-time applications. Nevertheless, the MBDS Adams software has been used herein for validation purposes.

The paper is divided into the following sections. Section 2 gives the fundamentals of the curved beam model. Section 3 outlines the kinetostatic analysis. Two simplified linear and partial non-linear models are developed, and their tangent stiffness matrices are obtained in closed form. Section 4 includes a detailed case study description. Section 5 compares and validates the two models and gives essential insights into convergence and computational burden. Finally, Section 6 gives the concluding remarks.

## 2. The Adopted Curved Beam Model

Flexures employed in this context are curved beams. It has been demonstrated that curved beams can provide large rotations while maintaining small errors in terms of displacements of its center [39], as it is typical for classic revolute pairs. This feature is important to guarantee finite rotations of the end-effector in monolithic structures such as MEMS-based grippers. Furthermore, a linear model is capable of faithfully reproducing the displacements and in-plane rotation of the curved beam tip up to rotations of approximately $\pm 20^{\circ }$. This feature has the considerable advantage of using a constant stiffness matrix, as will be recalled below.

In the following, the curved beam compliance matrix, and its inverse stiffness matrix, will be recalled from [36]. Let us consider a curved beam with a circular profile of radius $r_{f}$ and beam characteristic angle $\theta_{f}$, as displayed in Figure 1. First, let us consider the generalized displacement array $\psi_{f}={\left[{\hat{\xi }}_{f}^{T},\phi_{f}\right]}^{T}$ containing the displacement ${\hat{\xi }}_{f}$ and the rotation angle $\phi_{f}$ of the end section due to the deformation. Then, introducing the generalized wrench array $w_{f}={\left[F_{f}^{T},M_{f}\right]}^{T}$ containing the force vector and the torque applied to the end section, the compliance matrix $C_{f}$, derived in [36], follows from
(1)$$\psi_{f}=C_{f}w_{f}$$
and depends only on the geometric and structural parameters of the curved beam, i.e.,
(2)$$C_{f}=\frac{1}{EI}\left[\begin{array}{ccc} \frac{r_{f}^{3}}{4}\left(6\theta_{f}+s\left(2\theta_{f}\right)-8s\left(\theta_{f}\right)\right) & \frac{r_{f}^{3}}{2}\left(c^{2}\left(\theta_{f}\right)-2c\left(\theta_{f}\right)+1\right) & r_{f}^{2}(\theta_{f}-s\left(\theta_{f}\right)\\ \cdots & \frac{r_{f}^{3}}{4}\left(2\theta_{f}-s\left(2\theta_{f}\right)\right) & r_{f}^{2}\left(1-c\left(\theta_{f}\right)\right)\\ \;\left(\mathrm{sym}\right) & \cdots & r_{f}\theta_{f} \end{array}\right]$$
where *E* is Young’s modulus and *I* is the area moment of inertia, assumed both constant for the circular profile. In the following sections, the inverse of the compliance matrix $C_{f}$, i.e., the stiffness matrix $K_{f}$ will be employed to write the kinetostatic equations of planar mechanisms with curved beams. Furthermore, the stiffness matrix will be expressed in its locale frame ${\hat{S}}_{f}$ attached to the end-section of the curved beam in the undeformed configuration, as shown in Figure 1.

## 3. Kinetostatic Analysis

Hereafter, all vectors denoted with the *hat* will refer to the undeformed configuration, while the same vectors will indicate the deformed configuration without the *hat*. Position vectors of local frames as well as rotation matrices of frames describing the orientation of bodies in the undeformed configuration are constant.

A curved beam links two components, as it happens for the two bodies displayed in Figure 2. First, consider the undeformed system composed of two rigid bodies, identified by the reference frames ${\hat{S}}_{i}$ and ${\hat{S}}_{j}$, and by the flexure $\hat{f}$. The vectors ${\hat{r}}_{i}$ and ${\hat{r}}_{j}$ denote the positions of the body reference frame origins with respect to the fixed reference frame Σ. In contrast, the vectors ${\hat{s}}_{if}$ and ${\hat{s}}_{jf}$, respectively, indicate the distance vectors going from the body-reference frame origins to the attachment points of the curved beam to the bodies.

In the undeformed configuration, the position vector ${\hat{p}}_{f}$ going from the attachment point on body *i* to that on body *j* is obtained through the following expression
(3)$${\hat{p}}_{f}={\hat{r}}_{j}+{\hat{s}}_{jf}-{\hat{r}}_{i}-{\hat{s}}_{if}$$

Then, consider a generic configuration in which the bodies undergo finite displacements and rotations, and the flexure is deformed. For the assumption of rigid bodies, it follows that ${\hat{s}}_{if}^{\left({\hat{S}}_{i}\right)}\equiv s_{if}^{\left(S_{i}\right)}\equiv {\bar{s}}_{if}$ and ${\hat{s}}_{jf}^{\left({\hat{S}}_{j}\right)}\equiv s_{jf}^{\left(S_{j}\right)}\equiv {\bar{s}}_{jf}$ where the superscript denotes the reference frame in which the vector is expressed. In the previous expressions, ${\bar{s}}_{if}$ and ${\bar{s}}_{jf}$ have been introduced to simplify the notation. Then, the following closure equation stands,
(4)$$r_{i}+A_{i}{\bar{s}}_{if}+p_{f}-r_{j}-A_{j}{\bar{s}}_{jf}=0$$
where $A_{i}$ and $A_{j}$ are the rotation matrices mapping $S_{i}$ and $S_{j}$ into Σ and $p_{f}$ is the distance vector between the two flexure extremities in the deformed configuration.

Let us introduce the deformation vector $x_{f}^{\prime }$ containing the deformations of the flexure due to the displaced configuration described in Section 2. As known from the continuum mechanics, this vector can be represented using either the material or the spatial description of motion. In the following, only the material description is implemented. Therefore, expressing $x_{f}^{\prime }$ in the frame ${\hat{S}}_{if}$ of Figure 2, it follows
(5)$$x_{f}^{\prime }={\hat{A}}_{i}A_{i}^{T}p_{f}-{\hat{p}}_{f}$$
where $p_{f}$ has been pulled back to the undeformed configuration as required in the material description of motion.

Then, as recalled in Section 2, the circular flexure model requires $x_{f}^{\prime }$ to be expressed in the frame ${\hat{S}}_{fj}$ instead of ${\hat{S}}_{if}$, therefore
(6)$$x_{f}^{\prime ({\hat{S}}_{fj})}={\hat{A}}_{fj}^{T}{\hat{A}}_{j}^{T}x_{f}^{\prime }$$
where ${\hat{A}}_{fj}$ is the constant rotation matrix mapping ${\hat{S}}_{fj}$ into ${\hat{S}}_{j}$. The rotation angle $\phi_{f}$ due to the flexure deformation reads
(7)$$\phi_{f}=\theta_{j}-\theta_{i}-{\hat{\theta }}_{j}+{\hat{\theta }}_{i}\equiv \Delta \theta_{ij}-\Delta {\hat{\theta }}_{ij}$$
where θ and $\hat{\theta }$, respectively, are the rotation angles of the bodies in the spatial and material configurations and Δθ, $\Delta \hat{\theta }$ denote the corresponding relative rotation angles. The generalized displacement array of the curved beam *f*, already introduced in Section 2, becomes
(8)$$\psi_{f}^{\left({\hat{S}}_{fj}\right)}=\left[\begin{array}{c} x_{f}^{\prime ({\hat{S}}_{fj})}\\ \phi_{f} \end{array}\right]$$

### 3.1. Jacobian of the Deformation Vector

Since the direct kinetostatic analysis will be solved using an iterative procedure, the variation of the generalized displacement array must be calculated. Using Equation (Equation 5), the variation $\delta x_{f}^{\prime }$ is
(9)$$\delta x_{f}^{\prime }={\hat{A}}_{i}{\bar{A}}_{i}^{T}p_{f}\delta \theta_{i}+{\hat{A}}_{i}A_{i}^{T}\left(\delta r_{j}-\delta r_{i}+{\bar{A}}_{j}{\bar{s}}_{jf}\delta \theta_{j}-{\bar{A}}_{i}{\bar{s}}_{if}\delta \theta_{i}\right)$$
where $\bar{A}=\partial A/\partial \theta$ while δr, δθ are the variations of the body coordinates in the deformed configuration. The variation $\delta x_{f}^{\prime }$ is evaluated in the reference frame Σ but can be easily expressed in ${\hat{S}}_{fj}$ remembering that ${\hat{A}}_{fj}$ and ${\hat{A}}_{j}$ in Equation (Equation 6) are constant, i.e.,
(10)$$\delta x_{f}^{\prime ({\hat{S}}_{fj})}={\hat{A}}_{fj}^{T}{\hat{A}}_{j}^{T}\delta x_{f}^{\prime }$$

Considering the angles, the variation of Equation (Equation 7), leads to
(11)$$\delta \phi_{f}=\delta \theta_{j}-\delta \theta_{i}\equiv \delta \Delta \theta_{ij}$$

Finally, the variations can be combined to form the variation of the generalized deformation vector $\delta \psi_{f}$. The latter satisfies the following expression
(12)$$\delta \psi_{f}={\overset{˘}{J}}_{f}\delta q_{ij}$$
where $\delta q_{ij}={\left[\delta q_{i}^{T}\delta q_{j}^{T}\right]}^{T}$ is a 6-dimensional vector containing the variations of the body coordinates of bodies *i* and *j* and ${\overset{˘}{J}}_{f}$ is the (3×6) Jacobian matrix, defined as
(13)$${\overset{˘}{J}}_{f}=\left[{\overset{˘}{J}}_{f}^{i},|,{\overset{˘}{J}}_{f}^{j}\right]=\left[\begin{array}{cc} -{\hat{A}}_{i}A_{i}^{T} & {\hat{A}}_{i}\left({\bar{A}}_{i}^{T}p_{f}-\tilde{1}{\bar{s}}_{if}\right)\\ 0^{T} & -1 \end{array}|\begin{array}{cc} {\hat{A}}_{i}A_{i}^{T} & {\hat{A}}_{i}A_{i}^{T}{\bar{A}}_{j}{\bar{s}}_{jf}\\ 0^{T} & 1 \end{array}\right]$$

In deriving ${\overset{˘}{J}}_{f}$, the property $A_{i}^{T}{\bar{A}}_{i}=\tilde{1}$ has been employed, being
(14)$$\tilde{1}=\left[\begin{array}{cc} 0 & -1\\ 1 & 0 \end{array}\right]$$
a particular skew-symmetric matrix used to define the cross-product in the planar case.

### 3.2. Kinetostatic Equations

The kinetostatic equations of the system require the static equilibrium of a curved beam. Consider the layout of Figure 3 showing the static balance of a curved beam connecting the bodies *i* and *j*. The deformation of the beam yields force and moment applied on the section ${\hat{S}}_{fj}$ that must be equilibrated at section ${\hat{S}}_{if}$. A first simplified model, hereafter referred to as the *linear model*, performs the balance in the undeformed configuration and leads to the following expressions
(15)$$\left[\begin{array}{c} F_{if}^{\left({\hat{S}}_{if}\right)}\\ M_{if} \end{array}\right]=\underset{T_{f}}{\underset{︸}{-\left[\begin{array}{cc} A_{f} & 0\\ -{\left(A_{f}^{T}d_{f}^{\left({\hat{S}}_{if}\right)}\right)}^{T}\tilde{1} & 1 \end{array}\right]}}\left[\begin{array}{c} F_{fj}^{\left({\hat{S}}_{fj}\right)}\\ M_{fj} \end{array}\right]$$
where $A_{f}$ is the rotation matrix mapping ${\hat{S}}_{fj}$ to ${\hat{S}}_{if}$ and $d_{f}$ is the distance vector between the two sections, respectively, defined as
(16)$$A_{f}=\left[\begin{array}{cc} \cos(\theta_{f}) & -\sin(\theta_{f})\\ \sin(\theta_{f}) & \cos(\theta_{f}) \end{array}\right],\;d_{f}^{\left({\hat{S}}_{if}\right)}=\left[\begin{array}{c} r_{f}\sin\left(\theta_{f}\right)\\ r_{f}\left(1-\cos\left(\theta_{f}\right)\right) \end{array}\right]$$

In Section 2, the stiffness matrix of the curved beam has been derived considering the undeformed configuration, meaning that the stiffness model is linear and cannot capture the geometrical non-linearity coming from the change of configuration during the beam deformation. Despite this, the balance in the deformed configuration can be modified including the tip displacement due to deformation. Referring to Figure 3, the tip displacement can be included in deriving the moment $M_{if}$ at the first section ${\hat{S}}_{if}$, therefore modifying the previous Equation (Equation 15) into
(17)$$\left[\begin{array}{c} F_{if}^{\left({\hat{S}}_{if}\right)}\\ M_{if} \end{array}\right]=\underset{T_{f}^{\prime }}{\underset{︸}{-\left[\begin{array}{cc} A_{f} & 0\\ -{\left(A_{f}^{T}d_{f}^{\left({\hat{S}}_{if}\right)}+x_{f}^{\prime ({\hat{S}}_{fj})}\right)}^{T}\tilde{1} & 1 \end{array}\right]}}\left[\begin{array}{c} F_{fj}^{\prime ({\hat{S}}_{fj})}\\ M_{fj}^{\prime } \end{array}\right]$$
where $F_{fj}^{\prime }$ and $M_{fj}^{\prime }$ are referred to the deformed configuration. It is noteworthy that this *partial non-linear model* is not the geometrically exact fully non-linear model of the curved beam since the forces and moment at section ${\hat{S}}_{fj}$ are still obtained using a linear stiffness model for the curved beam.

In the following, either the linear or the partial non-linear model will be included to derive the kinetostatic equations of a planar mechanism. Let us consider the body *i* in its deformed configuration, as displayed in Figure 4. The flexures have been removed and replaced with their reaction forces and torques where the minus signs come from Newton’s third law. The static balance of body *i* requires that the following system be satisfied
(18a)$$\begin{array}{ccc} F_{i}-F_{1i}-F_{i2} & = & 0 \end{array}$$
(18b)$$\begin{array}{ccc} M_{i}-M_{1i}-M_{i2}+s_{i1}^{T}\tilde{1}F_{1i}+s_{i2}^{T}\tilde{1}F_{i2} & = & 0 \end{array}$$
where $F_{i}$ and $M_{i}$ are the external force and torque applied to body *i*, respectively. From the balance Equations (Equation 15) and (Equation 17), the forces and moments coming from the flexures have been expressed in the undeformed configuration and are now turned into the deformed one. From Figure 4, it can be found that $A_{1i}F_{1i}^{\left(S_{1i}\right)}\equiv {\hat{A}}_{1i}{\hat{F}}_{1i}^{\left({\hat{S}}_{1i}\right)}$ and $A_{i2}F_{i2}^{\left(S_{i2}\right)}\equiv {\hat{A}}_{i2}{\hat{F}}_{i2}^{\left({\hat{S}}_{i2}\right)}$, hence it follows that
(19a)$$\begin{array}{ccc} F_{i}-A_{i}{\hat{A}}_{1i}{\hat{F}}_{1i}^{\left({\hat{S}}_{1i}\right)}-A_{i}{\hat{A}}_{i2}{\hat{F}}_{i2}^{\left({\hat{S}}_{i2}\right)} & = & 0 \end{array}$$
(19b)$$\begin{array}{ccc} M_{i}-M_{1i}-M_{i2}+s_{i1}^{T}\tilde{1}F_{1i}+s_{i2}^{T}\tilde{1}F_{i2} & = & 0 \end{array}$$

Considering the frame invariance of the scalar equation of moments, the final system reads
(20a)$$\begin{array}{ccc} F_{i}-A_{i}{\hat{A}}_{1i}{\hat{F}}_{1i}^{\left({\hat{S}}_{1i}\right)}-A_{i}{\hat{A}}_{i2}{\hat{F}}_{i2}^{\left({\hat{S}}_{i2}\right)} & = & 0 \end{array}$$
(20b)$$\begin{array}{ccc} M_{i}-{\hat{M}}_{1i}-{\hat{M}}_{i2}+{\hat{s}}_{i1}^{T}\tilde{1}{\hat{A}}_{1i}{\hat{F}}_{1i}^{\left({\hat{S}}_{1i}\right)}+{\hat{s}}_{i2}^{T}\tilde{1}{\hat{A}}_{i2}{\hat{F}}_{i2}^{\left({\hat{S}}_{i2}\right)} & = & 0 \end{array}$$
or in matrix form
(21)$$\left[\begin{array}{c} F_{i}\\ M_{i} \end{array}\right]-\left[\begin{array}{cc} A_{i}{\hat{A}}_{1i} & 0\\ -{\hat{s}}_{i1}^{T}\tilde{1}{\hat{A}}_{1i} & 1 \end{array}\right]\left[\begin{array}{c} {\hat{F}}_{1i}^{\left({\hat{S}}_{1i}\right)}\\ {\hat{M}}_{1i} \end{array}\right]-\left[\begin{array}{cc} A_{i}{\hat{A}}_{i2} & 0\\ -{\hat{s}}_{i2}^{T}\tilde{1}{\hat{A}}_{i2} & 1 \end{array}\right]\left[\begin{array}{c} {\hat{F}}_{i2}^{\left({\hat{S}}_{i2}\right)}\\ {\hat{M}}_{i2} \end{array}\right]=0$$ Then, denoting with
(22)$$N_{1i}=\left[\begin{array}{cc} A_{i}{\hat{A}}_{1i} & 0\\ -{\hat{s}}_{i1}^{T}\tilde{1}{\hat{A}}_{1i} & 1 \end{array}\right],\;N_{i2}=\left[\begin{array}{cc} A_{i}{\hat{A}}_{i2} & 0\\ -{\hat{s}}_{i2}^{T}\tilde{1}{\hat{A}}_{i2} & 1 \end{array}\right]$$
and with $R_{i}$ the residual vector of body *i*, the final kinetostatic model is
(23)$$R_{i}\equiv -w_{i}+N_{1i}K_{1}^{\left({\hat{S}}_{1i}\right)}\psi_{1}^{\left({\hat{S}}_{1i}\right)}+N_{i2}T_{2}^{\prime }K_{2}^{\left({\hat{S}}_{2j}\right)}\psi_{2}^{\left({\hat{S}}_{2j}\right)}=0$$
where $w_{i}={\left[F_{i}^{T},M_{i}\right]}^{T}$ is the generalized force vector or wrench acting on body *i*, $K_{1}$ and $K_{2}$ are the stiffness matrices of the curved beams connected to the body, and $T_{2}^{\prime }$ is the matrix defined in Equation (Equation 17). If the latter is replaced by the matrix $T_{2}$ of the linear model in Equation (Equation 15), the kinetostatic model turns into
(24)$$R_{i}\equiv -w_{i}+N_{1i}K_{1}^{\left({\hat{S}}_{1i}\right)}\psi_{1}^{\left({\hat{S}}_{1i}\right)}+N_{i2}T_{2}K_{2}^{\left({\hat{S}}_{2j}\right)}\psi_{2}^{\left({\hat{S}}_{2j}\right)}=0$$

The kinetostatic equations of a complex multibody system are derived by assembling the residual vectors of all rigid bodies. The final system can be cast in the form
(25)R(w,q)=0,
where R indicates the global residual vector, w is the global wrench of external forces and torques, and q is the global vector of generalized coordinates. The elastostatics model offers two types of analyses: the inverse and the direct kinetostatic analysis. The deformed configuration is the input, and the global wrench is the output in the inverse analysis, as has been already described in [36]. The solution of the inverse kinetostatic analysis is straightforward and does not require an iterative procedure. In the direct kinetostatic analysis, the forces and moments applied to the system are known, while the final configuration of the deformed mechanism is sought. This highly non-linear problem can be solved using an iterative procedure such as the Newton–Raphson method described in Algorithm 1.
**Algorithm 1** Newton–Raphson iterative method1:ϵ← given threshold to terminate iterations2:k=1← iteration number3:$q^{\left(k\right)}=\hat{q}\leftarrow$ set the solution guess value4:**procedure**Newton–Raphson iterative method($q^{\left(k\right)}$,$\hat{q}$,w)5:    $R(w,q^{\left(k\right)})\leftarrow$ calculate the residuals as in Equation (Equation 24)6:    $K_{T}\left(q^{\left(k\right)}\right)\leftarrow$ calculate the tangent stiffness matrix as in Equation (Equation 26)7:    $q^{\left(k+1\right)}=q^{\left(k\right)}-{K_{T}^{\left(k\right)}}^{-1}R\left(w,q^{\left(k\right)}\right)\leftarrow$ update the solution8:    **if** $∥q^{\left(k+1\right)}-q^{\left(k\right)}∥<\epsilon$ **then**9:        $q=q^{\left(k+1\right)}\leftarrow$ DKP solution10:        exit **procedure**11:    **else**12:        k←k+113:        **goto** step 414:    **end if**15:**end procedure**

### 3.3. Tangent Stiffness Matrix Determination

Suppose that the external forces and moments are fixed in space. Then, considering the residual of the partial non-linear model of Equation (Equation 23), the tangent stiffness matrix is
(26)$$K_{Ti}=\frac{\partial R_{i}}{\partial q}\equiv K_{Ti}^{I}+K_{Ti}^{II}+K_{Ti}^{III}$$
where
(27a)$$\begin{array}{ccc} K_{Ti}^{I} & = & N_{1i}K_{1}^{\left({\hat{S}}_{1i}\right)}\frac{\partial \psi_{1}^{\left({\hat{S}}_{1i}\right)}}{\partial q}+N_{i2}T_{2}^{\prime }K_{2}^{\left({\hat{S}}_{2j}\right)}\frac{\partial \psi_{2}^{\left({\hat{S}}_{2j}\right)}}{\partial q} \end{array}$$
(27b)$$\begin{array}{ccc} K_{Ti}^{II} & = & \frac{\partial N_{1i}}{\partial q}K_{1}^{\left({\hat{S}}_{1i}\right)}\psi_{1}^{\left({\hat{S}}_{1i}\right)}+\frac{\partial N_{i2}}{\partial q}T_{2}^{\prime }K_{2}^{\left({\hat{S}}_{2j}\right)}\psi_{2}^{\left({\hat{S}}_{2j}\right)} \end{array}$$
(27c)$$\begin{array}{ccc} K_{Ti}^{III} & = & N_{i2}\frac{\partial T_{2}^{\prime }}{\partial q}K_{2}^{\left({\hat{S}}_{2j}\right)}\psi_{2}^{\left({\hat{S}}_{2j}\right)} \end{array}$$

If the residual of the linear model of Equation (Equation 24) is used instead, the tangent stiffness matrix turns into
(28)$$K_{Ti}=\frac{\partial R_{i}}{\partial q}\equiv K_{Ti}^{I}+K_{Ti}^{II}$$ The Appendix A reports the expressions for the terms of $K_{Ti}$.

## 4. Case Study: The Four-Bar Linkage

In this section, a compliant gripper with curved beams, shown in Figure 5a, is studied. The monolithic structure of the mechanism can be reduced to two in-parallel four-bar linkages, as revealed in Figure 5b. For symmetry along the vertical axis, only half a mechanism will be analyzed, i.e., the right side of Figure 5b. The layout of Figure 6 has been plotted following the notation presented in Section 3. Body 1 is the frame, here considered fixed, while bodies 2, 3, and 4 are moving rigid bodies. Considering grippers with CSFH joints, one link is a part of the monolithic structure enclosed between two circular beams. In an MEMS device, the entire monolithic structure can deform. However, the assumption of rigid links coupled to flexible circular beams has been verified using FE models. It is fully justified since the links are at least one order of magnitude stiffer than the circular beam flexures.

First, vectors ${\hat{p}}_{f}$ can be calculated knowing the undeformed configuration, i.e., $\hat{q}$, therefore
(29a)$$\begin{array}{ccc} {\hat{p}}_{a} & = & {\hat{r}}_{2}-{\hat{r}}_{1}+{\hat{A}}_{2}{\hat{s}}_{2a}-{\hat{A}}_{1}{\hat{s}}_{1a} \end{array}$$
(29b)$$\begin{array}{ccc} {\hat{p}}_{b} & = & {\hat{r}}_{3}-{\hat{r}}_{2}+{\hat{A}}_{3}{\hat{s}}_{3b}-{\hat{A}}_{2}{\hat{s}}_{2b} \end{array}$$
(29c)$$\begin{array}{ccc} {\hat{p}}_{c} & = & {\hat{r}}_{4}-{\hat{r}}_{3}+{\hat{A}}_{4}{\hat{s}}_{4c}-{\hat{A}}_{3}{\hat{s}}_{3c} \end{array}$$
(29d)$$\begin{array}{ccc} {\hat{p}}_{d} & = & {\hat{r}}_{1}-{\hat{r}}_{4}+{\hat{A}}_{1}{\hat{s}}_{1d}-{\hat{A}}_{4}{\hat{s}}_{4d} \end{array}$$

Similarly, the vectors $p_{f}$ of the flexures in the deformed configuration, i.e., q, are
(30a)$$\begin{array}{ccc} p_{a} & = & r_{2}-r_{1}+A_{2}s_{2a}-A_{1}s_{1a} \end{array}$$
(30b)$$\begin{array}{ccc} p_{b} & = & r_{3}-r_{2}+A_{3}s_{3b}-A_{2}s_{2b} \end{array}$$
(30c)$$\begin{array}{ccc} p_{c} & = & r_{4}-r_{3}+A_{4}s_{4c}-A_{3}s_{3c} \end{array}$$
(30d)$$\begin{array}{ccc} p_{d} & = & r_{1}-r_{4}+A_{1}s_{1d}-A_{4}s_{4d} \end{array}$$
Since body 1 is the frame, $r_{1}={\hat{r}}_{1}$ and $A_{1}={\hat{A}}_{1}$. Furthermore, if the frame $S_{1}$ of body 1 is coincident with Σ, it follows that $r_{1}=0$ and $A_{1}=1$. Here, these matrices are written for the sake of completeness.

Notice that q could be one of the iterative solutions $q^{\left(k\right)}$ employed in the Newton–Raphson algorithm. The deformation vectors $x_{f}^{\prime }$ in the material description and expressed in the local frames of the undeformed flexures are
(31a)$$\begin{array}{ccc} x_{a}^{\prime ({\hat{S}}_{a2})} & = & {\hat{A}}_{a2}^{T}{\hat{A}}_{2}^{T}\left({\hat{A}}_{1}A_{1}^{T}p_{a}-{\hat{p}}_{a}\right) \end{array}$$
(31b)$$\begin{array}{ccc} x_{b}^{\prime ({\hat{S}}_{b3})} & = & {\hat{A}}_{b3}^{T}{\hat{A}}_{3}^{T}\left({\hat{A}}_{2}A_{2}^{T}p_{b}-{\hat{p}}_{b}\right) \end{array}$$
(31c)$$\begin{array}{ccc} x_{c}^{\prime ({\hat{S}}_{c4})} & = & {\hat{A}}_{c4}^{T}{\hat{A}}_{4}^{T}\left({\hat{A}}_{3}A_{3}^{T}p_{c}-{\hat{p}}_{c}\right) \end{array}$$
(31d)$$\begin{array}{ccc} x_{d}^{\prime ({\hat{S}}_{d1})} & = & {\hat{A}}_{d1}^{T}{\hat{A}}_{1}^{T}\left({\hat{A}}_{4}A_{4}^{T}p_{d}-{\hat{p}}_{d}\right) \end{array}$$
The angular deformation $\phi_{f}$ is obtained as
(32a)$$\begin{array}{ccc} \phi_{a} & = & \theta_{2}-\theta_{1}-{\hat{\theta }}_{2}+{\hat{\theta }}_{1} \end{array}$$
(32b)$$\begin{array}{ccc} \phi_{b} & = & \theta_{3}-\theta_{2}-{\hat{\theta }}_{3}+{\hat{\theta }}_{2} \end{array}$$
(32c)$$\begin{array}{ccc} \phi_{c} & = & \theta_{4}-\theta_{3}-{\hat{\theta }}_{4}+{\hat{\theta }}_{3} \end{array}$$
(32d)$$\begin{array}{ccc} \phi_{d} & = & \theta_{1}-\theta_{4}-{\hat{\theta }}_{1}+{\hat{\theta }}_{4} \end{array}$$
The expressions (31) and (32) allows for determining the flexure generalized deformations $\psi_{a}^{\left({\hat{S}}_{a2}\right)}$, $\psi_{b}^{\left({\hat{S}}_{b3}\right)}$, $\psi_{c}^{\left({\hat{S}}_{c4}\right)}$, and $\psi_{d}^{\left({\hat{S}}_{d1}\right)}$.

The transformation matrices $T_{f}^{\prime }$ of Equation (Equation 17) are defined as
(33a)$$\begin{array}{ccc} T_{a}^{\prime } & = & -\left[\begin{array}{cc} A_{a} & 0\\ -{\left(A_{a}^{T}d_{a}^{\left({\hat{S}}_{1a}\right)}+x_{a}^{\prime ({\hat{S}}_{a2})}\right)}^{T}\tilde{1} & 0 \end{array}\right] \end{array}$$
(33b)$$\begin{array}{ccc} T_{b}^{\prime } & = & -\left[\begin{array}{cc} A_{b} & 0\\ -{\left(A_{b}^{T}d_{b}^{\left({\hat{S}}_{2b}\right)}+x_{b}^{\prime ({\hat{S}}_{b3})}\right)}^{T}\tilde{1} & 0 \end{array}\right] \end{array}$$
(33c)$$\begin{array}{ccc} T_{c}^{\prime } & = & -\left[\begin{array}{cc} A_{c} & 0\\ -{\left(A_{c}^{T}d_{c}^{\left({\hat{S}}_{3c}\right)}+x_{c}^{\prime ({\hat{S}}_{c4})}\right)}^{T}\tilde{1} & 0 \end{array}\right] \end{array}$$
(33d)$$\begin{array}{ccc} T_{d}^{\prime } & = & -\left[\begin{array}{cc} A_{d} & 0\\ -{\left(A_{d}^{T}d_{d}^{\left({\hat{S}}_{4d}\right)}+x_{d}^{\prime ({\hat{S}}_{d1})}\right)}^{T}\tilde{1} & 0 \end{array}\right] \end{array}$$
where $A_{f}$ and $d_{f}$ can be found for each curved beam using Equation (Equation 16). Expressions similar to $T_{f}^{\prime }$, not reported for brevity, can be written to determine $T_{f}$ of Equation (Equation 15).

Then, the matrices N of Equation (Equation 22) are
(34a)$$N_{d1}=\left[\begin{array}{cc} A_{1}{\hat{A}}_{d1} & 0\\ -{\hat{s}}_{1d}^{T}\tilde{1}{\hat{A}}_{d1} & 1 \end{array}\right],\;N_{1a}=\left[\begin{array}{cc} A_{1}{\hat{A}}_{1a} & 0\\ -{\hat{s}}_{1a}^{T}\tilde{1}{\hat{A}}_{1a} & 1 \end{array}\right]$$
(34b)$$N_{a2}=\left[\begin{array}{cc} A_{2}{\hat{A}}_{a2} & 0\\ -{\hat{s}}_{2a}^{T}\tilde{1}{\hat{A}}_{a2} & 1 \end{array}\right],\;N_{2b}=\left[\begin{array}{cc} A_{2}{\hat{A}}_{2b} & 0\\ -{\hat{s}}_{2b}^{T}\tilde{1}{\hat{A}}_{2b} & 1 \end{array}\right]$$
(34c)$$N_{b3}=\left[\begin{array}{cc} A_{3}{\hat{A}}_{b3} & 0\\ -{\hat{s}}_{3b}^{T}\tilde{1}{\hat{A}}_{b3} & 1 \end{array}\right],\;N_{3c}=\left[\begin{array}{cc} A_{3}{\hat{A}}_{3c} & 0\\ -{\hat{s}}_{3c}^{T}\tilde{1}{\hat{A}}_{3c} & 1 \end{array}\right]$$
(34d)$$N_{c4}=\left[\begin{array}{cc} A_{4}{\hat{A}}_{c4} & 0\\ -{\hat{s}}_{4c}^{T}\tilde{1}{\hat{A}}_{c4} & 1 \end{array}\right],\;N_{4d}=\left[\begin{array}{cc} A_{4}{\hat{A}}_{4d} & 0\\ -{\hat{s}}_{4d}^{T}\tilde{1}{\hat{A}}_{4d} & 1 \end{array}\right]$$

Setting the external wrenches $w_{i}$ applied at the mass centers $G_{i}$ of the rigid bodies, the four residual vectors $R_{i}$, i=1,…,4, are
(35a)$$\begin{array}{ccc} R_{1} & \equiv & -w_{1}+N_{d1}K_{d}^{\left({\hat{S}}_{d1}\right)}\psi_{d}^{\left({\hat{S}}_{d1}\right)}+N_{1a}T_{a}^{\prime }K_{a}^{\left({\hat{S}}_{a2}\right)}\psi_{a}^{\left({\hat{S}}_{a2}\right)} \end{array}$$
(35b)$$\begin{array}{ccc} R_{2} & \equiv & -w_{2}+N_{a2}K_{a}^{\left({\hat{S}}_{a2}\right)}\psi_{a}^{\left({\hat{S}}_{a2}\right)}+N_{2b}T_{b}^{\prime }K_{b}^{\left({\hat{S}}_{b3}\right)}\psi_{b}^{\left({\hat{S}}_{b3}\right)} \end{array}$$
(35c)$$\begin{array}{ccc} R_{3} & \equiv & -w_{3}+N_{b3}K_{b}^{\left({\hat{S}}_{b3}\right)}\psi_{b}^{\left({\hat{S}}_{b3}\right)}+N_{3c}T_{c}^{\prime }K_{c}^{\left({\hat{S}}_{c4}\right)}\psi_{c}^{\left({\hat{S}}_{c4}\right)} \end{array}$$
(35d)$$\begin{array}{ccc} R_{4} & \equiv & -w_{4}+N_{c4}K_{c}^{\left({\hat{S}}_{c4}\right)}\psi_{c}^{\left({\hat{S}}_{c4}\right)}+N_{4d}T_{d}^{\prime }K_{d}^{\left({\hat{S}}_{d1}\right)}\psi_{d}^{\left({\hat{S}}_{d1}\right)} \end{array}$$
The residuals are employed to form a system of 12 non-linear kinetostatic equations, i.e.,
(36)$$R\equiv {\left[R_{1}^{T},R_{2}^{T},R_{3}^{T},R_{4}^{T}\right]}^{T}=0$$
that must be solved using an iterative procedure. To calculate the tangent stiffness matrix necessary to apply the Newton–Raphson algorithm, let us define the Jacobians ${\overset{˘}{J}}_{f}^{\left({\hat{S}}_{fj}\right)}$, i.e.,
(37a)$${\overset{˘}{J}}_{a}^{\left({\hat{S}}_{a2}\right)}\equiv [\;{\overset{˘}{J}}_{a}^{1({\hat{S}}_{a2})}\;|\;{\overset{˘}{J}}_{a}^{2({\hat{S}}_{a2})}\;]=\left[\begin{array}{cc} {\hat{A}}_{a2}^{T}{\hat{A}}_{2}^{T} & 0\\ 0^{T} & 1 \end{array}\right]{\overset{˘}{J}}_{a}$$
(37b)$${\overset{˘}{J}}_{b}^{\left({\hat{S}}_{b3}\right)}\equiv [\;{\overset{˘}{J}}_{b}^{2({\hat{S}}_{b3})}\;|\;{\overset{˘}{J}}_{b}^{3({\hat{S}}_{b3})}\;]=[\begin{array}{cc} {\hat{A}}_{b3}^{T}{\hat{A}}_{3}^{T} & 0\\ 0^{T} & 1 \end{array}]{\overset{˘}{J}}_{b}$$
(37c)$${\overset{˘}{J}}_{c}^{\left({\hat{S}}_{c4}\right)}\equiv [\;{\overset{˘}{J}}_{c}^{3({\hat{S}}_{c4})}\;|\;{\overset{˘}{J}}_{c}^{4({\hat{S}}_{c4})}\;]=[\begin{array}{cc} {\hat{A}}_{c4}^{T}{\hat{A}}_{4}^{T} & 0\\ 0^{T} & 1 \end{array}]{\overset{˘}{J}}_{c}$$
(37d)$${\overset{˘}{J}}_{d}^{\left({\hat{S}}_{d1}\right)}\equiv [\;{\overset{˘}{J}}_{d}^{4({\hat{S}}_{d1})}\;|\;{\overset{˘}{J}}_{d}^{1({\hat{S}}_{d1})}\;]=[\begin{array}{cc} {\hat{A}}_{d1}^{T}{\hat{A}}_{1}^{T} & 0\\ 0^{T} & 1 \end{array}]{\overset{˘}{J}}_{d}$$
where
(38a)$${\overset{˘}{J}}_{a}=\left[\begin{array}{cc} -{\hat{A}}_{1}A_{1}^{T} & {\hat{A}}_{1}\left({\bar{A}}_{1}^{T}p_{a}-\tilde{1}{\bar{s}}_{1a}\right)\\ 0^{T} & -1 \end{array}|\begin{array}{cc} {\hat{A}}_{1}A_{1}^{T} & {\hat{A}}_{1}A_{1}^{T}{\bar{A}}_{2}{\bar{s}}_{2a}\\ 0^{T} & 1 \end{array}\right]$$
(38b)$${\overset{˘}{J}}_{b}=\left[\begin{array}{cc} -{\hat{A}}_{2}A_{2}^{T} & {\hat{A}}_{2}\left({\bar{A}}_{2}^{T}p_{b}-\tilde{1}{\bar{s}}_{2b}\right)\\ 0^{T} & -1 \end{array}|\begin{array}{cc} {\hat{A}}_{2}A_{2}^{T} & {\hat{A}}_{2}A_{2}^{T}{\bar{A}}_{3}{\bar{s}}_{3b}\\ 0^{T} & 1 \end{array}\right]$$
(38c)$${\overset{˘}{J}}_{c}=\left[\begin{array}{cc} -{\hat{A}}_{3}A_{3}^{T} & {\hat{A}}_{1}\left({\bar{A}}_{3}^{T}p_{c}-\tilde{1}{\bar{s}}_{3c}\right)\\ 0^{T} & -1 \end{array}|\begin{array}{cc} {\hat{A}}_{3}A_{3}^{T} & {\hat{A}}_{3}A_{3}^{T}{\bar{A}}_{4}{\bar{s}}_{4c}\\ 0^{T} & 1 \end{array}\right]$$
(38d)$${\overset{˘}{J}}_{d}=\left[\begin{array}{cc} -{\hat{A}}_{4}A_{4}^{T} & {\hat{A}}_{4}\left({\bar{A}}_{4}^{T}p_{d}-\tilde{1}{\bar{s}}_{4d}\right)\\ 0^{T} & -1 \end{array}|\begin{array}{cc} {\hat{A}}_{4}A_{4}^{T} & {\hat{A}}_{4}A_{4}^{T}{\bar{A}}_{1}{\bar{s}}_{1d}\\ 0^{T} & 1 \end{array}\right]$$
The Jacobians ${\overset{˘}{J}}_{f}^{\left({\hat{S}}_{fj}\right)}$ can be mapped using Boolean matrices to the final dimension of the system, i.e.,
(39a)$$J_{a}^{\left({\hat{S}}_{a2}\right)}=[\;{\overset{˘}{J}}_{a}^{1({\hat{S}}_{a2})}\;|\;{\overset{˘}{J}}_{a}^{2({\hat{S}}_{a2})}\;|\;O\;|\;O]$$
(39b)$$J_{b}^{\left({\hat{S}}_{b3}\right)}=[\;O\;|\;{\overset{˘}{J}}_{b}^{2({\hat{S}}_{b3})}\;|\;{\overset{˘}{J}}_{b}^{3({\hat{S}}_{b3})}\;|\;O\;]$$
(39c)$$J_{c}^{\left({\hat{S}}_{c4}\right)}=[\;O\;|\;O\;|\;{\overset{˘}{J}}_{c}^{3({\hat{S}}_{c4})}\;|\;{\overset{˘}{J}}_{c}^{4({\hat{S}}_{c4})}\;]$$
(39d)$$J_{d}^{\left({\hat{S}}_{d1}\right)}=[\;{\overset{˘}{J}}_{d}^{1({\hat{S}}_{d1})}\;|\;O\;|\;O\;|\;{\overset{˘}{J}}_{d}^{4({\hat{S}}_{d1})}\;]$$
Then, following the expression (Equation 50) of $K_{Ti}^{I}$ reported in the Appendix, it yields
(40a)$$\begin{array}{ccc} K_{T1}^{I} & = & N_{d1}K_{d}^{\left({\hat{S}}_{d1}\right)}J_{d}^{\left({\hat{S}}_{d1}\right)}+N_{1a}T_{a}^{\prime }K_{a}^{\left({\hat{S}}_{a2}\right)}J_{a}^{\left({\hat{S}}_{a2}\right)} \end{array}$$
(40b)$$\begin{array}{ccc} K_{T2}^{I} & = & N_{a2}K_{a}^{\left({\hat{S}}_{a2}\right)}J_{a}^{\left({\hat{S}}_{a2}\right)}+N_{2b}T_{b}^{\prime }K_{b}^{\left({\hat{S}}_{b3}\right)}J_{b}^{\left({\hat{S}}_{b3}\right)} \end{array}$$
(40c)$$\begin{array}{ccc} K_{T3}^{I} & = & N_{b3}K_{b}^{\left({\hat{S}}_{b3}\right)}J_{b}^{\left({\hat{S}}_{b3}\right)}+N_{3c}T_{c}^{\prime }K_{c}^{\left({\hat{S}}_{c4}\right)}J_{c}^{\left({\hat{S}}_{c4}\right)} \end{array}$$
(40d)$$\begin{array}{ccc} K_{T4}^{I} & = & N_{c4}K_{c}^{\left({\hat{S}}_{c4}\right)}J_{c}^{\left({\hat{S}}_{c4}\right)}+N_{4d}T_{d}^{\prime }K_{d}^{\left({\hat{S}}_{d1}\right)}J_{d}^{\left({\hat{S}}_{d1}\right)} \end{array}$$

The final expression for the first part of the tangent stiffness matrix is
(41)$$K_{T}^{I}=\left[\begin{array}{c} K_{T1}\\ K_{T2}^{I}\\ K_{T3}^{I}\\ K_{T4}^{I} \end{array}\right]$$

The expressions for $K_{Ti}^{II}$ can be obtained starting from $z_{i}$ in Equation (Equation 55), i.e.,
(42a)$$\begin{array}{ccc} z_{1} & = & G_{d1}K_{d}^{\left({\hat{S}}_{d1}\right)}\psi_{d}^{\left({\hat{S}}_{d1}\right)}+G_{1a}T_{a}^{\prime }K_{a}^{\left({\hat{S}}_{a2}\right)}\psi_{a}^{\left({\hat{S}}_{a2}\right)} \end{array}$$
(42b)$$\begin{array}{ccc} z_{2} & = & G_{a2}K_{a}^{\left({\hat{S}}_{a2}\right)}\psi_{a}^{\left({\hat{S}}_{a2}\right)}+G_{2b}T_{b}^{\prime }K_{b}^{\left({\hat{S}}_{b3}\right)}\psi_{b}^{\left({\hat{S}}_{b3}\right)} \end{array}$$
(42c)$$\begin{array}{ccc} z_{3} & = & G_{b3}K_{b}^{\left({\hat{S}}_{b3}\right)}\psi_{b}^{\left({\hat{S}}_{b3}\right)}+G_{3c}T_{c}^{\prime }K_{c}^{\left({\hat{S}}_{c3}\right)}\psi_{c}^{\left({\hat{S}}_{c4}\right)} \end{array}$$
(42d)$$\begin{array}{ccc} z_{4} & = & G_{c4}K_{c}^{\left({\hat{S}}_{c4}\right)}\psi_{c}^{\left({\hat{S}}_{c4}\right)}+G_{2b}T_{d}^{\prime }K_{d}^{\left({\hat{S}}_{d4}\right)}\psi_{d}^{\left({\hat{S}}_{d4}\right)} \end{array}$$
where the matrices G of Equation (Equation 52) are defined as
(43a)$$G_{d1}=\left[\begin{array}{cc} {\bar{A}}_{1}{\hat{A}}_{d1} & 0\\ 0^{T} & 1 \end{array}\right],\;G_{1a}=\left[\begin{array}{cc} {\bar{A}}_{1}{\hat{A}}_{1a} & 0\\ 0^{T} & 1 \end{array}\right]$$
(43b)$$G_{a2}=\left[\begin{array}{cc} {\bar{A}}_{2}{\hat{A}}_{a2} & 0\\ 0^{T} & 1 \end{array}\right],\;G_{2b}=\left[\begin{array}{cc} {\bar{A}}_{2}{\hat{A}}_{2b} & 0\\ 0^{T} & 1 \end{array}\right]$$
(43c)$$G_{b3}=\left[\begin{array}{cc} {\bar{A}}_{3}{\hat{A}}_{b3} & 0\\ 0^{T} & 1 \end{array}\right],\;G_{3c}=\left[\begin{array}{cc} {\bar{A}}_{3}{\hat{A}}_{3c} & 0\\ 0^{T} & 1 \end{array}\right]$$
(43d)$$G_{c4}=\left[\begin{array}{cc} {\bar{A}}_{4}{\hat{A}}_{c4} & 0\\ 0^{T} & 1 \end{array}\right],\;G_{4d}=\left[\begin{array}{cc} {\bar{A}}_{4}{\hat{A}}_{4d} & 0\\ 0^{T} & 1 \end{array}\right]$$
Therefore, $K_{T}^{II}$ becomes
(44)$$K_{T}^{II}=\left[\begin{array}{ccc} 0 & 0 & z_{1}\\ 0 & 0 & 0\\ 0 & 0 & 0\\ 0 & 0 & 0 \end{array}|\begin{array}{ccc} 0 & 0 & 0\\ 0 & 0 & z_{2}\\ 0 & 0 & 0\\ 0 & 0 & 0 \end{array}|\begin{array}{ccc} 0 & 0 & 0\\ 0 & 0 & 0\\ 0 & 0 & z_{3}\\ 0 & 0 & 0 \end{array}|\begin{array}{ccc} 0 & 0 & 0\\ 0 & 0 & 0\\ 0 & 0 & 0\\ 0 & 0 & z_{4} \end{array}\right]$$

Finally, the third part of the tangent stiffness matrix $K_{T}^{III}$ is derived through the 6-dimensional vectors $v_{f}$ of Equation (Equation 59), i.e.,
(45a)$$\begin{array}{ccc} v_{a} & = & {\overset{˘}{J}}_{ax}^{T}{\hat{A}}_{2}{\hat{A}}_{a2}\tilde{1}F_{a}^{\left({\hat{S}}_{a2}\right)}\equiv J_{ax}^{({\hat{S}}_{a2})T}\tilde{1}F_{a}^{\left({\hat{S}}_{a2}\right)} \end{array}$$
(45b)$$\begin{array}{ccc} v_{b} & = & {\overset{˘}{J}}_{bx}^{T}{\hat{A}}_{3}{\hat{A}}_{b3}\tilde{1}F_{b}^{\left({\hat{S}}_{b3}\right)}\equiv J_{bx}^{({\hat{S}}_{b3})T}\tilde{1}F_{b}^{\left({\hat{S}}_{b3}\right)} \end{array}$$
(45c)$$\begin{array}{ccc} v_{c} & = & {\overset{˘}{J}}_{cx}^{T}{\hat{A}}_{4}{\hat{A}}_{c4}\tilde{1}F_{c}^{\left({\hat{S}}_{c4}\right)}\equiv J_{cx}^{({\hat{S}}_{c4})T}\tilde{1}F_{c}^{\left({\hat{S}}_{c4}\right)} \end{array}$$
(45d)$$\begin{array}{ccc} v_{d} & = & {\overset{˘}{J}}_{dx}^{T}{\hat{A}}_{1}{\hat{A}}_{d1}\tilde{1}F_{d}^{\left({\hat{S}}_{d1}\right)}\equiv J_{dx}^{({\hat{S}}_{d1})T}\tilde{1}F_{d}^{\left({\hat{S}}_{d1}\right)} \end{array}$$
where $F_{f}^{\left({\hat{S}}_{fj}\right)}$ is obtained taking the force vector from the flexure wrench $w_{f}^{\left({\hat{S}}_{fj}\right)}=K_{f}^{\left({\hat{S}}_{fj}\right)}\psi_{f}^{\left({\hat{S}}_{fj}\right)}$. The block-matrices ${\overset{˘}{J}}_{fx}$, or $J_{fx}^{\left({\hat{S}}_{fj}\right)}$, are derived taking only the first two rows, i.e., the block of $x_{f}^{\prime }$, from the corresponding Jacobian matrices. Using the equations of (Equation 60), the matrices $V_{f}=\left[\begin{array}{ccc} V_{fi} & | & V_{fj} \end{array}\right]$ can be obtained and therefore, the matrix $K_{T}^{III}$ becomes
(46)$$K_{T}^{III}=\left[\begin{array}{c} N_{1a}V_{a1}\\ O\\ O\\ N_{4d}V_{d1} \end{array}|\begin{array}{c} N_{1a}V_{a2}\\ N_{2b}V_{b2}\\ O\\ O \end{array}|\begin{array}{c} O\\ N_{2b}V_{b3}\\ N_{3c}V_{c3}\\ O \end{array}|\begin{array}{c} O\\ O\\ N_{3c}V_{c4}\\ N_{4d}V_{d4} \end{array}\right]$$
The final expression for the tangent stiffness matrix is $K_{T}=K_{T}^{I}+K_{T}^{II}+K_{T}^{III}$. If the DOFs of the first body, i.e., the fixed frame, are removed by imposing fixed boundary conditions, the final form of $K_{T}$ will be a (9×9) matrix with the following pattern
(47)$$K_{T}=\left[\begin{array}{ccccc} • & | & • & | & \\ & | & • & | & •\\ & | & & | & • \end{array}\right]$$

## 5. Numerical Application

Referring to Figure 6, body 1 is fixed while body 2 is actuated through a vertical force applied at its center of mass. Finally, the end-effector is attached to body 3. Following the layout of Figure 6, all geometric and structural parameters necessary for direct kinetostatic analysis of the case study are reported in Table 1.

### 5.1. Comparison and Validation

First, let us consider the linear flexure model expressed through Equation (Equation 24) of Section 3. Considering an initial actuation force $F_{2y}=-60$ (μN), the latter is increased until the *x*-coordinate of the end-effector becomes zero, i.e., the gripper clamp is completely closed. The gripper deforms as displayed on the left side of Figure 7, wherein the limit cases and the undeformed configuration are reported. Then, let us consider the partial non-linear flexure model expressed through the Equation (Equation 23) of Section 3. Performing the same simulations, the workspace becomes that of the right side of Figure 7. The values of $F_{2y}$ for which the clamp is closed, respectively, are $F_{2y}=63$ (μN) for the linear model and $F_{2y}=77$ (μN) for the partial non-linear model.

The two models have been compared to a model implemented using the commercial multibody software MSC Adams${}^{©}$. The model includes rigid links and flexible circular beams. The latter are combined to the links’ endpoints through fixed connections. The model has been developed to be comparable with the Matlab model. The only difference pertains to the flexures since the curved beams have been modeled using a two-dimensional geometrical non-linear representation for beam-like structures. Compared to the previous Matlab models, this representation is fully non-linear, as the stiffness matrix of each flexure is updated during deformation. The actuation force has been applied through step functions, and two simulations have been carried out to achieve convergency; one for the forward movement of closing and the other for the backward movement of opening. Figure 8 shows the two simulations and the trajectories accomplished by the end-effector point. As for Figure 7, the deformed configuration has also been plotted to understand the deformation process better.

The three models have been compared in Figure 9 in terms of the end-effector trajectory, also referred to as the mechanism’s workspace (left subplot) and actuation forces (right subplot).

Two materials have been modeled for the curved beams: silicon with Young modulus $E_{y}=100$ (GPa) and nylon with Young modulus $E_{y}=3.84$ (GPa). Considering the same rectangular cross-section, whose dimensions are reported in Table 1, the silicon bending stiffness is EI=26.0417 (μNμm${}^{2}$) while the nylon bending stiffness is EI=1 (μNμm${}^{2}$). The two materials have different properties in terms of elasticity. Silicon has a brittle behavior, while nylon has an anisotropic hyperelastic or visco-hyperelastic behavior. The two bending stiffness values should be seen as extreme cases to test the proposed method. With this premise in mind, the two materials will be assumed to have both isotropic linear behavior, while the different degrees of non-linearity of the models will only concern the geometric stiffness.

The first row of plots in Figure 9 pertains to the silicon while the second one is the nylon. First, let us consider silicon. Observing the top-left subplot, the three arc-shaped trajectories of the end-effector reveal relevant differences only in the final part of the path. The influence of the fully non-linear flexures becomes more evident in the opening movement, where the trajectories become more distant. Compared to the linear model, the top-right subplot reveals that the flexure non-linearities introduce a stiffening effect, and the actuation force required to produce the same displacement grows. It can be observed that the partial non-linear model is stiffer than the fully non-linear model in the forward closing movement and softer in the opening movement. The three models have equal stiffness only at the undeformed configuration where the actuation force is zero.

Then, let us consider nylon. Observing the bottom-left subplot, the end-effector trajectory follows a trend similar to the previous case. The bottom-right plot reveals differences in force range, as could be expected considering the lower bending stiffness of the nylon. Now, the non-linearities are more pronounced, and three inflection points appear for the fully non-linear flexures, which are totally absent in the remaining cases. Despite this, the plot is similar to the simplified cases.

Excluding the limit points of the workspace during the opening movement, the simplified models provide excellent results. Added to this is that the proper workspace is usually limited by other constraints such as the electrical interfaces or the maximum stress in the material, thus making the three models closer than they might appear in Figure 9.

For example, the Conjugate Surfaces Flexure Hinges (CSFH) employed in MEMS micro-grippers have a rotation range limited to ±20${}^{\circ }$ to prevent the silicon from breaking [39,40]. This range is displayed in the opening–closing movement of Figure 9. However, this range is further limited to about ±2${}^{\circ }$ by other phenomena coming from the electrostatic actuation such as sticking-friction anomalies, the *pull-in* or the impossibility to generating high actuation forces [41].

### 5.2. Shape Optimization

The simplified models have been employed to perform the cross-section optimization of the curved beams, as shown in Figure 10. These plots can be used in various ways; for example, knowing the maximum actuation force the electrical interface can produce, the section parameters can be chosen to cope with this value. Another example could be related to the choice of the section parameters based on the maximum allowable stress of the material or its fatigue limit. Likewise, the optimization could affect other structural parameters or mechanism lengths.

### 5.3. Tangent Stiffness Matrix

Since the tangent stiffness matrix is the key element of the direct kinetostatic analysis, in the following, a detailed analysis of the tangent stiffness matrix and its role in the Newton–Raphson algorithm convergence is detailed. As already described in Section 4, after imposing fixed boundary conditions on body 1, the tangent stiffness matrix turns into a 9×9 symmetric matrix. Referring to the partial non-linear model, $K_{T}$ has the expression reported in Table 2 in the undeformed configuration.

Let us consider the mechanism in the final deformed configuration obtained by applying $F_{2y}=50$ (μN). In this case, the expression of $K_{T}$ is reported in Table 3, while the percentage difference between the deformed and undeformed case is provided in Table 4. It can be observed that relevant differences appear during the deformation process, especially in the off-diagonal components. Furthermore, the matrix $K_{T}$ is no longer symmetrical in the deformed configuration.

Since $K_{T}$ is composed of three terms, it is legitimate to ask what the contribution of each term is. As reported in Table 5, the first term $K_{T}^{I}$ is the closest to the final expression of $K_{T}$.

The second term $K_{T}^{II}$ is reported in Table 6. Remembering the expression for $z_{i}$ in Equation (Equation 55) and the form of $K_{T}^{II}$ in Equation (Equation 56), observing Table 6, it can be found that at the equilibrium, i.e., if and only if the residuals are zero, the following expression stands
(48)$$z_{i}=\left[\begin{array}{cc} \tilde{1} & 0\\ 0^{T} & 0 \end{array}\right],\;w_{i}\equiv \left[\begin{array}{c} -F_{iy}\\ +F_{ix}\\ 0 \end{array}\right]$$ For the case study, only body 2, and therefore $z_{2}$, has components different from zero.

Finally, the third term $K_{T}^{III}$ is reported in Table 7. It can be noticed that only the components of the inner moments, i.e., due to the flexures, are different from zero. This feature comes from the particular form of $V_{fi}$, or $V_{fj}$, in Equation (Equation 60).

To better focus on the importance of the tangent stiffness matrix in achieving solution convergence, let us keep the tangent stiffness matrix constant and equal to that obtained in the undeformed configuration for the entire simulation, i.e., $K_{T}=K_{T}\left(\hat{q}\right)$. The results of Figure 11 reveal that the number of iterations necessary to achieve convergence grows exponentially as the input load increases and no longer converges beyond $F_{2y}=76$ (μN).

The same simulation has been repeated by updating the tangent stiffness matrix following four strategies based on:The complete matrix $K_{T}$ starting each simulation from the undeformed solution;The first term $K_{T}^{I}$ starting each simulation from the undeformed solution;The complete matrix $K_{T}$ starting each simulation from the previous converged solution;The first term $K_{T}^{I}$ starting each simulation from the previous converged solution.

It can be seen that the approaches using the previous converged solution reduce the number of iterations. Similarly, using the complete matrix instead of the first-term approximated matrix results in fewer iterations. The results are displayed in Figure 12.

It is interesting to observe how these trends translate into a computational burden. The reduced number of iterations provided by the strategies based on the previous converged solution translates directly into savings in computation time. The two strategies based on the first term of the tangent stiffness matrix lead to a higher number of iterations but, simultaneously, require a smaller number of variables to be determined and save on calculation times, as shown in Figure 13. The peaks observed in Figure 13 come from memory allocation and other inner processes of Matlab during the first computation. On the other hand, it can be observed from Figure 12 that a higher number of iterations do not correspond to these CPU times. The results have been obtained using an HP workstation equipped with an Intel Xeon CPU @3.20GHz with 32 GB of RAM.

This section is concluded by giving some insights into the computational time obtained using Adams. In Table 8, the CPU times obtained in the opening–closing movement for the Adams fully non-linear model, the Matlab linear model, and the Matlab partial non-linear model are compared. Adams simulations have been performed by disabling the graphic display. The comparison has been carried out for both silicon and nylon. As can be observed, the CPU time of the simplified methods is from 30 to 500 times faster than Adams. It is noteworthy that the simulation time for nylon is ten times faster than silicon for the Adams fully non-linear model.

## 6. Conclusions and Discussion

The tangent stiffness matrix has been used as a conceptual base to solve the direct kinetostatic problem of planar grippers with curved beams. Two models have been presented to cope with flexure deformations. The first linear model considers the flexure equilibrium in the initial undeformed configuration, while the second partial non-linear model considers the equilibrium in the deformed configuration. Both methods do not include a fully non-linear geometric description of the curved beam flexure whose stiffness matrix is kept constant and equal to that obtained in the undeformed state. The tangent stiffness matrix has been divided into sub-parts to facilitate both the theoretical treatment and the numerical implementation. The linear model led to two sub-parts, while the partial non-linear model introduced a further third sub-part. Both models were tested and compared with a fully non-linear model obtained using the commercial software MSC Adams. The results proved to be in good agreement on most of the mechanism’s workspace, except for the extreme areas wherein the geometric non-linear effects become relevant. The same case study was used to show the method’s potential; for example, in conducting a shape optimization of the flexure cross-section. Finally, the importance of each term of the tangent stiffness matrix in the convergence process was detailed in terms of the number of iterations required to achieve convergence and computational load.

From what has been outlined, the proposed method offers various advantages:1.The results of Figure 13 suggest a possible extension to real-time applications of micro and nano-grippers. It is known that the control often requires simplified models to be executed quickly by the control unit. Often these models are obtained by linearizing the equilibrium equations around one or more operating points. Using models with reduced complexity would allow more efficient control strategies such as control in the operating space, inverse dynamics control, pre-calculated torque control. Furthermore, the closed form helps creating more efficient reduced order models [42,43,44].2.The tangent stiffness matrix is obtained in closed form. This feature prevents the use of numeric differentiation, making the convergence process of the direct kinetostatic solution more robust. Furthermore, splitting the expression of $K_{T}$ allowed for identifying its most basic terms and calibrating the compromise between the number of iterations and calculation time. The calculation times are considerably reduced by using only the first term of the tangent stiffness matrix and recalculating it at each iteration of the Newton–Raphson algorithm described in Algorithm 1.3.The tangent stiffness matrix can be employed to develop a dynamics model to study vibrations. The tangent stiffness matrix is the core of implicit time integration methods primarily employed in flexible multibody dynamics [45]. Shape optimization takes further advantage of the closed form of $K_{T}$ opening scenarios to gradient-based constrained optimization problems based on the kinetostatic analysis.4.Both the two simplified models employ curved beams modeled by a constant stiffness matrix. Despite this, the curved beams guarantee finite displacements/rotations in the mechanism, allowing for the expansion of the reachable workspace. The model remains reliable for most of the mechanism’s workspace. The results are accurate in the functional area except for the limit zones of the workspace in which physical constraints usually prevent motion. When the maximum rotations of the curved beams exceed $\pm 20^{\circ }$ the constant stiffness hypothesis can no longer capture the geometric nonlinearities, and the results deviate from the actual case.5.Although the proposed method is valid only for planar cases, it can be extended to other compliant mechanisms with constant stiffness flexures without changing the mathematical background.

## Figures and Tables

**Figure 1 micromachines-13-02172-f001:**
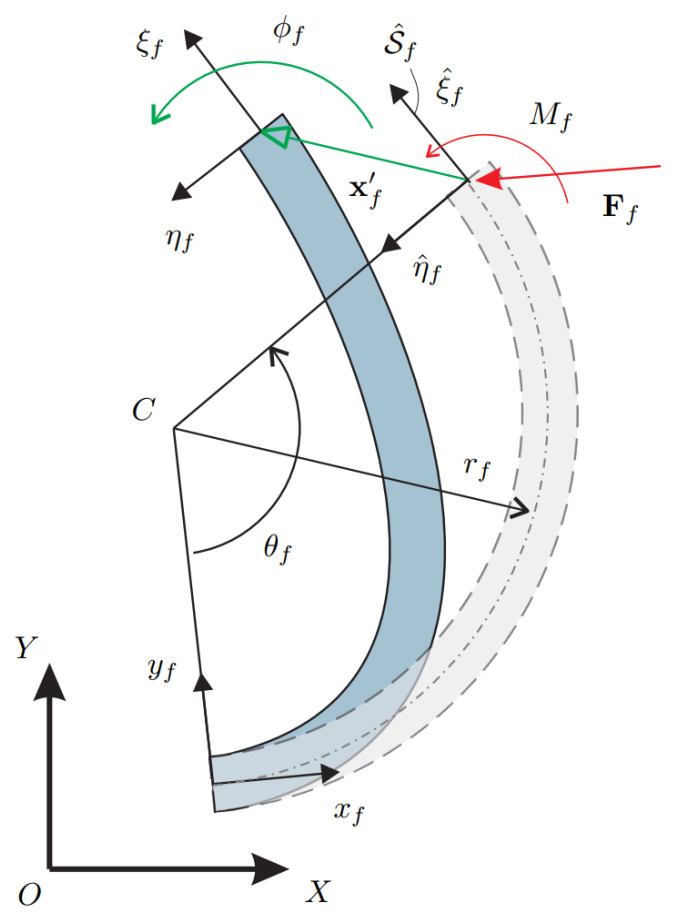
Curved beam in its initial (dashed line) and deformed configuration (solid line).

**Figure 2 micromachines-13-02172-f002:**
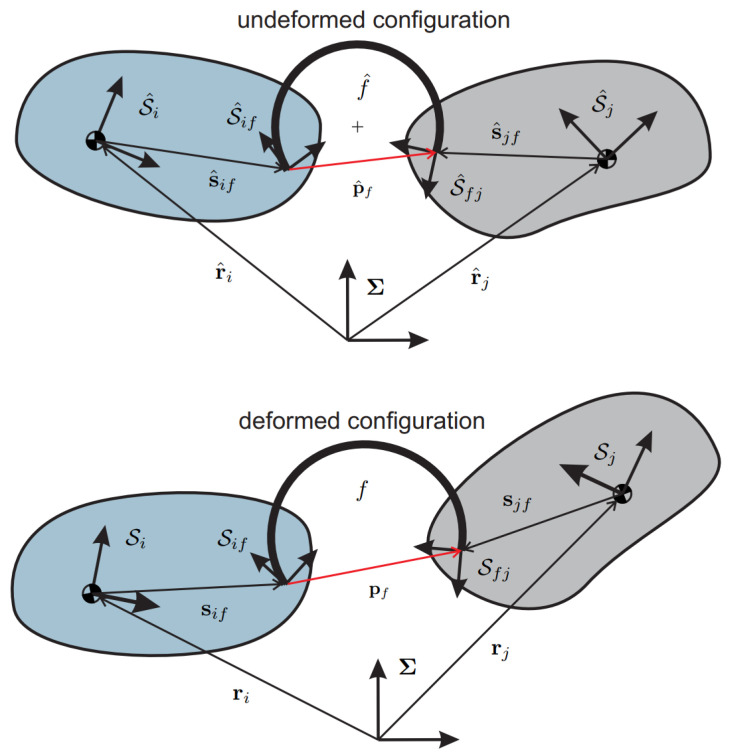
Deformation of a curved beam due to the relative motions of the bodies connected to its extremities.

**Figure 3 micromachines-13-02172-f003:**
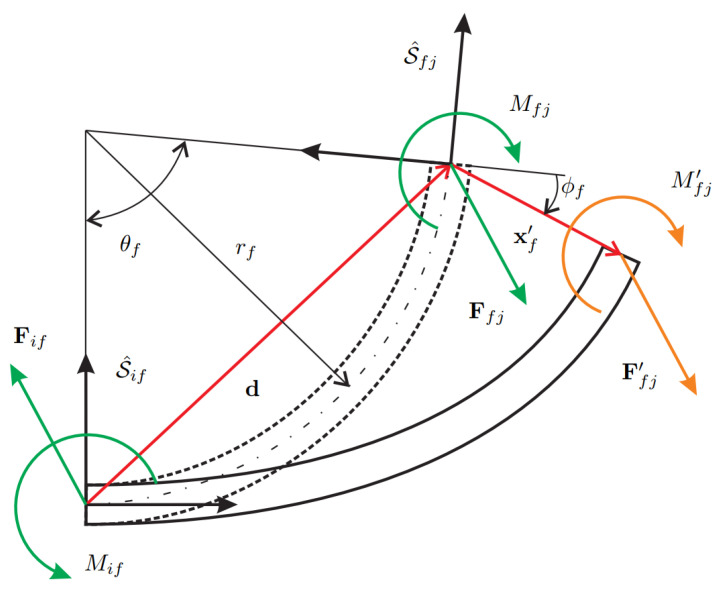
Static balance of a curved beam. Dashed line for the undeformed beam and a solid line for the deformed beam.

**Figure 4 micromachines-13-02172-f004:**
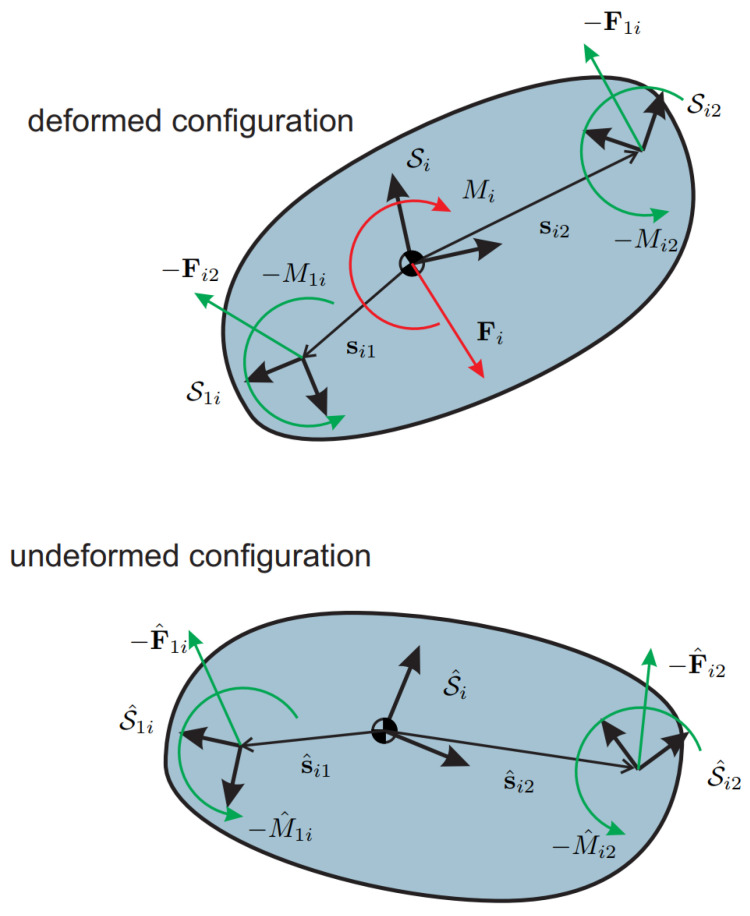
Kinetostatic balance of a rigid-body.

**Figure 5 micromachines-13-02172-f005:**
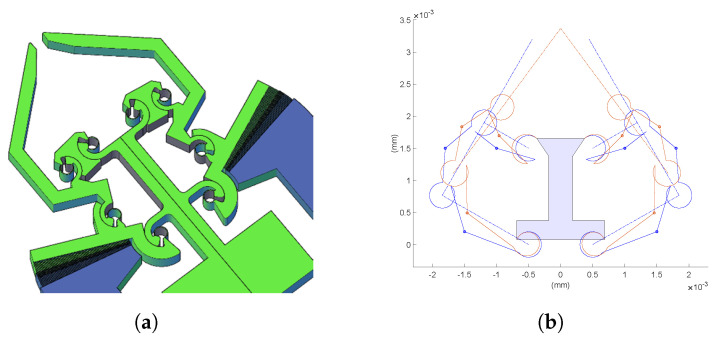
MEMS-based gripper with curved beams. (**a**) CAD layout. (**b**) Matlab model: (red) deformed configuration, (blue) undeformed configuration.

**Figure 6 micromachines-13-02172-f006:**
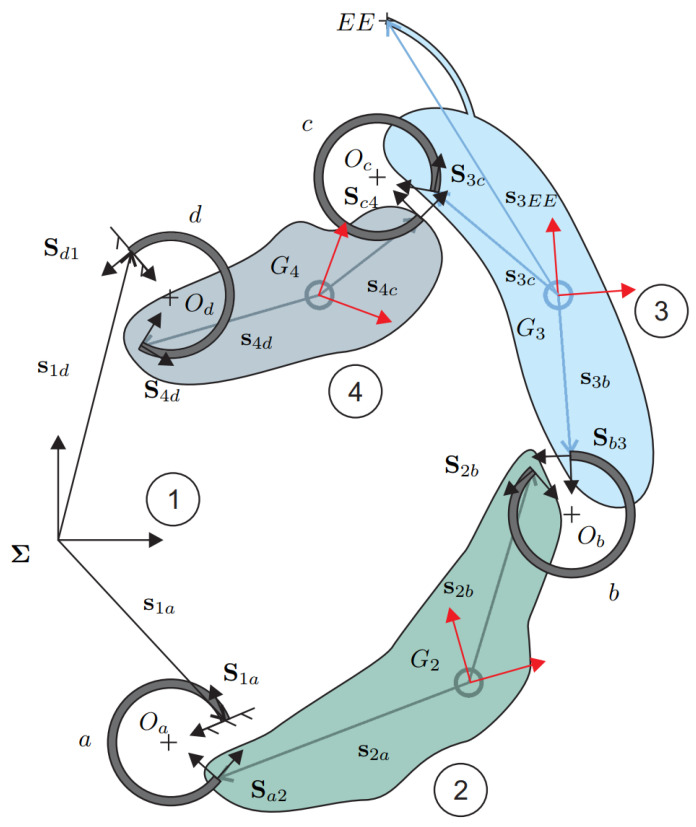
Layout of a four-bar linkage with curved beams.

**Figure 7 micromachines-13-02172-f007:**
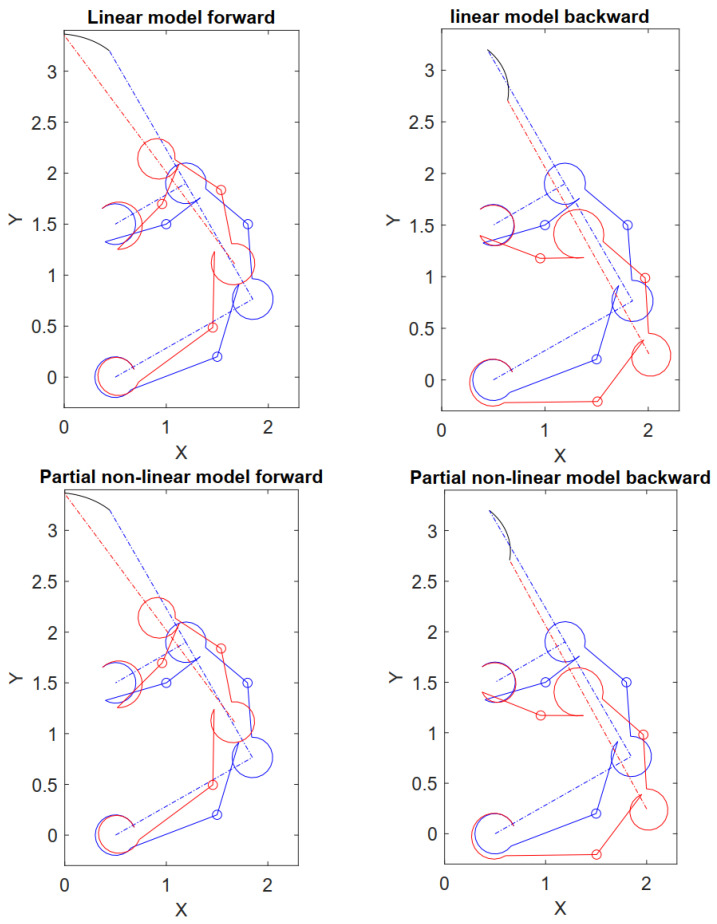
Workspace of the gripper and end-effector trajectory for the linear and partial non-linear models during the opening/closing maneuver: (red) deformed configuration, (blue) undeformed configuration, (black) end-effector trajectory. Units in millimeters.

**Figure 8 micromachines-13-02172-f008:**
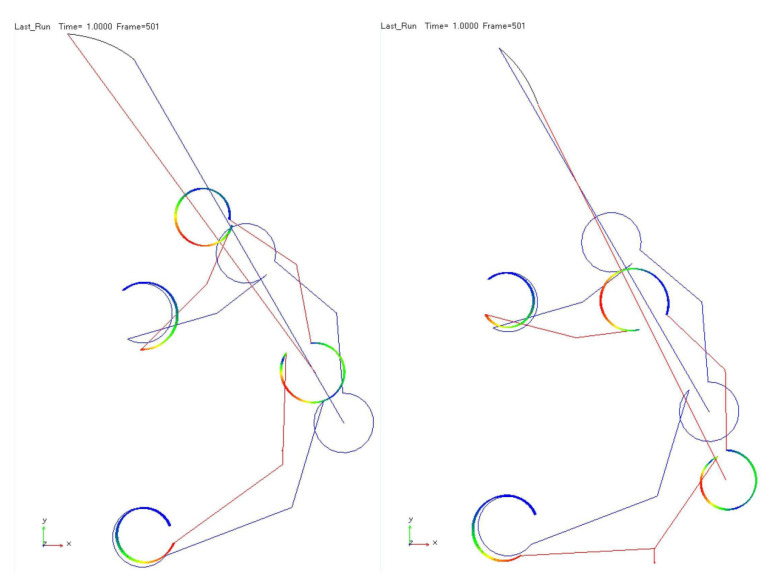
Adams results: (**left**) forward movement; (**right**) backward movement. The deformed gripper is in red, while the undeformed configuration is represented in blue color. The trajectory of the end-effector for the two movements is displayed in black color.

**Figure 9 micromachines-13-02172-f009:**
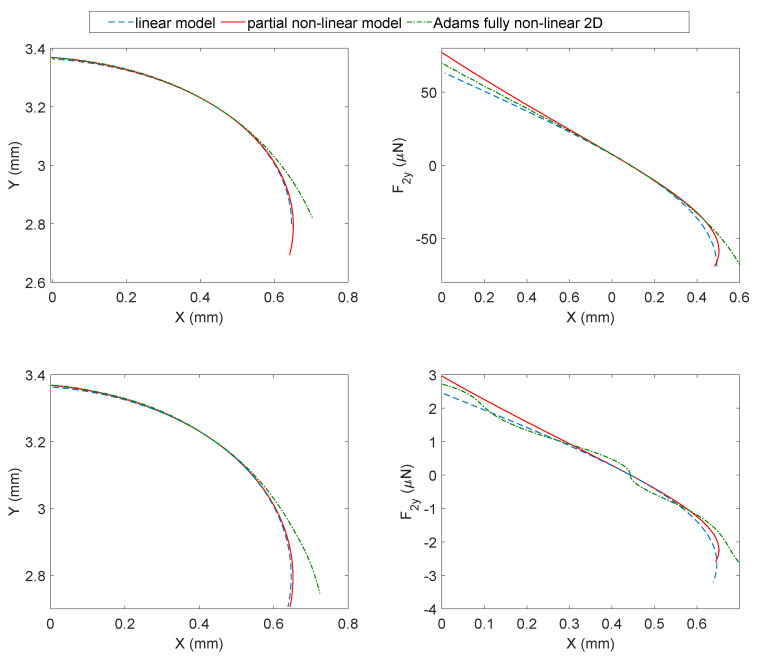
Model comparison in terms of workspace and actuation forces. Left subplot: end-effector trajectory in an opening–closing movement; right subplot: actuation force vs. *x*-coordinate of the end-effector during the opening–closing movement.

**Figure 10 micromachines-13-02172-f010:**
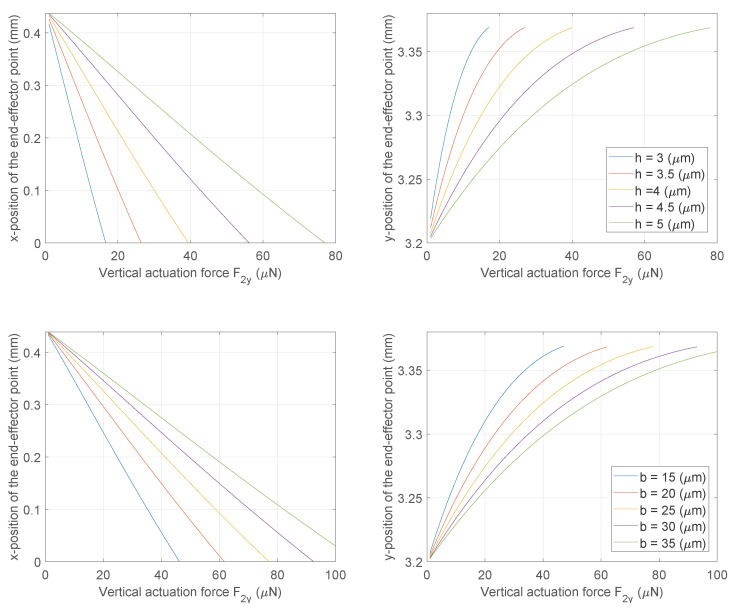
Cross-section optimization of the flexures.

**Figure 11 micromachines-13-02172-f011:**
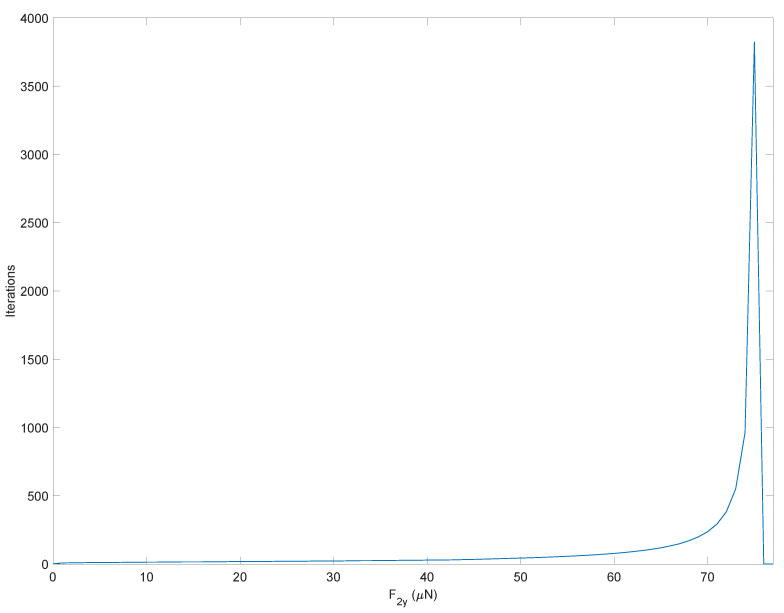
Number of iterations to achieve convergence of the direct kinetostatic analysis varying the input force. The simulation employs the constant tangent stiffness matrix obtained in the undeformed configuration.

**Figure 12 micromachines-13-02172-f012:**
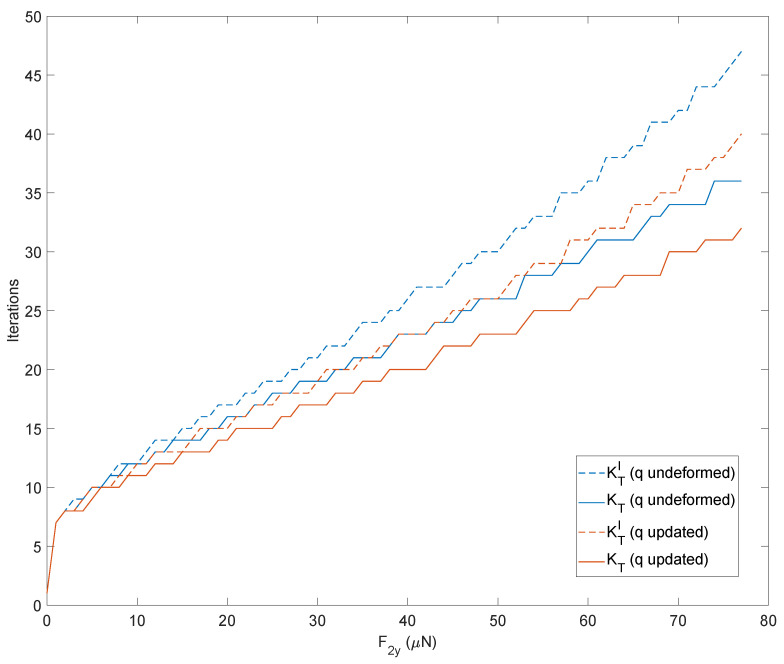
Number of iterations to achieve convergency of the direct kinetostatic analysis varying the input force. The simulation compares four strategies to upload the tangent stiffness matrix: considering the complete matrix $K_{T}$ or only the first term $K_{T}^{I}$ starting each simulation from the undeformed solution, considering the complete matrix $K_{T}$ or only the first term $K_{T}^{I}$ starting each simulation from the previous converged solution.

**Figure 13 micromachines-13-02172-f013:**
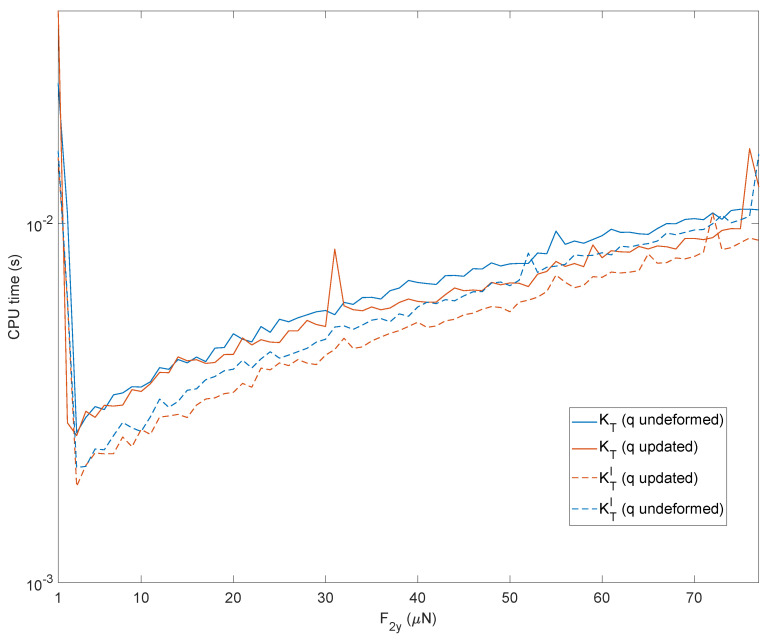
Computation time of the four strategies of Figure 12.

**Table 1 micromachines-13-02172-t001:** Geometric and structural parameters of the case study.

Four-Bar Mechanism		
mass center $G_{2}$	[1.5000, 0.2000]	(mm)
mass center $G_{3}$	[1.8000, 1.5000]	(mm)
mass center $G_{4}$	[1.0000, 1.5000]	(mm)
hinge center $O_{a}$	[0.5000, 0.0000]	(mm)
hinge center $O_{b}$	[1.8478, 0.7654]	(mm)
hinge center $O_{c}$	[1.1928, 1.9000]	(mm)
hinge center $O_{d}$	[0.5000, 1.5000]	(mm)
local base vector ${\bar{s}}_{1a}$	[0.6848, 0.0765]	(mm)
local base vector ${\bar{s}}_{2a}$	[−0.8413, −0.3218]	(mm)
local base vector ${\bar{s}}_{2b}$	[0.2139, 0.7140]	(mm)
local base vector ${\bar{s}}_{3b}$	[0.0408, −0.5348]	(mm)
local base vector ${\bar{s}}_{3c}$	[−0.4140, 0.3482]	(mm)
local base vector ${\bar{s}}_{3EE}$	[−1.3571, 1.6991]	(mm)
local base vector ${\bar{s}}_{4c}$	[0.3342, 0.2586]	(mm)
local base vector ${\bar{s}}_{4d}$	[−0.6000, −0.1732]	(mm)
local base vector ${\bar{s}}_{1d}$	[0.3714, 1.6532]	(mm)
curved beam		
beam radius $r_{f},\;\left(f\in \left[a,b,c,d\right]\right)$	0.2	(mm)
beam characteristic angle $\left[\theta_{a},\theta_{b},\theta_{c},\theta_{d}\right]$	[300, 320, 330, 250]	(${}^{\circ }$)
Young modulus $E_{y}$	100	(GPa)
cross-section base *b*	25	(μm)
cross-section height *h*	5	(μm)

**Table 2 micromachines-13-02172-t002:** Tangent stiffness matrix of the partial non-linear model in the undeformed configuration.

2.40 × $10^{3}$	1.84 × $10^{2}$	−5.42 ×$10^{2}$	−1.32 ×$10^{3}$	−1.11 ×$10^{2}$	−1.01 ×$10^{3}$			
1.84 × $10^{2}$	2.71 × $10^{3}$	−1.34 ×$10^{3}$	−1.11 ×$10^{2}$	−1.09 ×$10^{3}$	−1.46 ×$10^{2}$			
−5.42 ×$10^{2}$	−1.34 ×$10^{3}$	2.29 × $10^{3}$	6.76 × $10^{2}$	−3.29 ×$10^{2}$	4.70 × $10^{2}$			
−1.32 ×$10^{3}$	−1.11 ×$10^{2}$	6.76 × $10^{2}$	2.41 × $10^{3}$	2.06 × $10^{2}$	4.98 × $10^{2}$	−1.10 ×$10^{3}$	−9.53 ×$10^{1}$	4.31 × $10^{2}$
−1.11 ×$10^{2}$	−1.09 ×$10^{3}$	−3.29 ×$10^{2}$	2.06 × $10^{2}$	2.30 × $10^{3}$	−6.44 ×$10^{2}$	−9.53 ×$10^{1}$	−1.21 ×$10^{3}$	−1.75 ×$10^{2}$
−1.01 ×$10^{3}$	−1.46 ×$10^{2}$	4.70 × $10^{2}$	4.98 × $10^{2}$	−6.44 ×$10^{2}$	1.51 × $10^{3}$	5.07 × $10^{2}$	7.89 × $10^{2}$	−9.01 ×$10^{1}$
			−1.10 ×$10^{3}$	−9.53 ×$10^{1}$	5.07 × $10^{2}$	2.34 × $10^{3}$	−5.58 ×$10^{1}$	−3.75 ×$10^{2}$
			−9.53 ×$10^{1}$	−1.21 ×$10^{3}$	7.89 × $10^{2}$	−5.58 ×$10^{1}$	4.16 × $10^{3}$	−1.08 ×$10^{3}$
			4.31 × $10^{2}$	−1.75 ×$10^{2}$	−9.01 ×$10^{1}$	−3.75 ×$10^{2}$	−1.08 ×$10^{3}$	7.93 × $10^{2}$

**Table 3 micromachines-13-02172-t003:** Tangent stiffness matrix of the partial non-linear model in the final deformed configuration obtained applying $F_{2y}=50$ (μN).

2.31 × $10^{3}$	−9.15 ×$10^{1}$	−1.66 ×$10^{2}$	−1.27 ×$10^{3}$	−1.45 ×$10^{2}$	−9.67 ×$10^{2}$			
4.22 × $10^{2}$	2.75 × $10^{3}$	−1.20 ×$10^{3}$	−1.45 ×$10^{2}$	−1.14 ×$10^{3}$	−2.65 ×$10^{2}$			
−5.90 ×$10^{2}$	−1.45 ×$10^{3}$	2.20 × $10^{3}$	7.24 × $10^{2}$	−2.25 ×$10^{2}$	4.81 × $10^{2}$			
−1.28 ×$10^{3}$	−2.79 ×$10^{2}$	6.74 × $10^{2}$	2.36 × $10^{3}$	3.65 × $10^{2}$	5.15 × $10^{2}$	−1.08 ×$10^{3}$	−8.63 ×$10^{1}$	5.32 × $10^{2}$
6.01 × $10^{0}$	−1.11 ×$10^{3}$	−3.28 ×$10^{2}$	8.03 × $10^{1}$	2.33 × $10^{3}$	−6.07 ×$10^{2}$	−8.63 ×$10^{1}$	−1.22 ×$10^{3}$	−2.72 ×$10^{1}$
−9.59 ×$10^{2}$	−3.34 ×$10^{2}$	4.46 × $10^{2}$	4.64 × $10^{2}$	−4.42 ×$10^{2}$	1.46 × $10^{3}$	4.95 × $10^{2}$	7.76 × $10^{2}$	−2.18 ×$10^{2}$
			−1.02 ×$10^{3}$	2.52 × $10^{2}$	2.50 × $10^{2}$	2.23 × $10^{3}$	−1.41 ×$10^{3}$	−1.90 ×$10^{2}$
			−3.80 ×$10^{2}$	−1.20 ×$10^{3}$	8.58 × $10^{2}$	6.65 × $10^{2}$	3.92 × $10^{3}$	−1.12 ×$10^{3}$
			4.43 × $10^{2}$	−1.43 ×$10^{2}$	−1.12 ×$10^{2}$	−4.97 ×$10^{2}$	−9.28 ×$10^{2}$	6.76 × $10^{2}$

**Table 4 micromachines-13-02172-t004:** Percentage difference of the tangent stiffness matrices of the partial non-linear model between the final deformed configuration of Table 3 and the undeformed configuration of Table 2.

−3.5	−149.8	−69.4	−3.8	31.4	−3.8			
129.6	1.2	−10.4	31.4	4.5	82.0			
8.8	7.8	−4.1	7.1	−31.6	2.3			
−3.1	152.5	−0.4	−2.3	77.6	3.5	−1.2	−9.4	23.4
−105.4	2.2	−0.2	−61.0	1.6	−5.6	−9.4	1.1	−84.4
−4.6	129.3	−5.1	−6.7	−31.4	−3.9	−2.5	−1.7	141.6
			−7.2	−364.2	−50.7	−4.4	2424.5	−49.4
			299.1	−0.8	8.7	−1290.9	−5.8	3.7
			2.7	−18.2	23.9	32.7	−14.1	−14.7

**Table 5 micromachines-13-02172-t005:** The first term $K_{T}^{I}$ in the deformed configuration obtained applying $F_{2y}=50$ (μN).

2.31 × $10^{3}$	−9.15 ×$10^{1}$	−1.16 ×$10^{2}$	−1.27 ×$10^{3}$	−1.45 ×$10^{2}$	−9.67 ×$10^{2}$			
4.22 × $10^{2}$	2.75 × $10^{3}$	−1.20 ×$10^{3}$	−1.45 ×$10^{2}$	−1.14 ×$10^{3}$	−2.65 ×$10^{2}$			
−5.74 ×$10^{2}$	−1.42 ×$10^{3}$	2.19 × $10^{3}$	7.08 × $10^{2}$	−2.46 ×$10^{2}$	4.71 × $10^{2}$			
−1.28 ×$10^{3}$	−2.79 ×$10^{2}$	6.74 × $10^{2}$	2.36 × $10^{3}$	3.65 × $10^{2}$	5.15 × $10^{2}$	−1.08 ×$10^{3}$	−8.63 ×$10^{1}$	5.32 × $10^{2}$
6.01 × $10^{0}$	−1.11 ×$10^{3}$	−3.28 ×$10^{2}$	8.03 × $10^{1}$	2.33 × $10^{3}$	−6.07 ×$10^{2}$	−8.63 ×$10^{1}$	−1.22 ×$10^{3}$	−2.72 ×$10^{1}$
−9.59 ×$10^{2}$	−3.34 ×$10^{2}$	4.46 × $10^{2}$	4.83 × $10^{2}$	−4.22 ×$10^{2}$	1.44 × $10^{3}$	4.76 × $10^{2}$	7.56 × $10^{2}$	−2.15 ×$10^{2}$
			−1.02 ×$10^{3}$	2.52 × $10^{2}$	2.50 × $10^{2}$	2.23 × $10^{3}$	−1.41 ×$10^{3}$	−1.90 ×$10^{2}$
			−3.80 ×$10^{2}$	−1.20 ×$10^{3}$	8.58 × $10^{2}$	6.65 × $10^{2}$	3.92 × $10^{3}$	−1.12 ×$10^{3}$
			4.43 × $10^{2}$	−1.43 ×$10^{2}$	−1.12 ×$10^{2}$	−4.77 ×$10^{2}$	−9.10 ×$10^{2}$	6.62 × $10^{2}$

**Table 6 micromachines-13-02172-t006:** Second term $K_{T}^{II}$ in the deformed configuration obtained applying $F_{2y}=50$ (μN).

0	0	−5.00 ×$10^{1}$	0	0	0			
0	0	−6.25 ×$10^{-12}$	0	0	0			
0	0	0	0	0	0			
0	0	0	0	0	−6.75 ×$10^{-13}$	0	0	0
0	0	0	0	0	4.73 × $10^{-13}$	0	0	0
0	0	0	0	0	0	0	0	0
			0	0	0	0	0	−1.95 ×$10^{-11}$
			0	0	0	0	0	7.21 × $10^{-11}$
			0	0	0	0	0	0

**Table 7 micromachines-13-02172-t007:** Third term $K_{T}^{III}$ in the deformed configuration obtained applying $F_{2y}=50$ (μN).

0	0	0	0	0	0			
0	0	0	0	0	0			
−1.62 ×$10^{1}$	−2.14 ×$10^{1}$	8.03 × $10^{0}$	1.62 × $10^{1}$	−2.14 ×$10^{1}$	−1.03 ×$10^{1}$			
0	0	0	0	0	0	0	0	0
0	0	0	0	0	0	0	0	0
0	0	0	−1.86 ×$10^{1}$	−1.94 ×$10^{1}$	1.30 × $10^{1}$	1.86 × $10^{1}$	1.94 × $10^{1}$	−2.29 ×$10^{0}$
			0	0	0	0	0	0
			0	0	0	0	0	0
			0	0	0	−1.99 ×$10^{1}$	−1.80 ×$10^{1}$	1.45 × $10^{1}$

**Table 8 micromachines-13-02172-t008:** CPU-time comparison in seconds for the opening–closing movement.

Silicon		
	opening	closing
Adams fully non-linear model	41.00	50.00
Matlab linear model	0.150	0.100
Matlab partial non-linear model	0.150	0.150
**Nylon**		
	opening	closing
Adams Fully non-linear model	4.500	4.500
Matlab linear model	0.135	0.100
Matlab partial non-linear model	0.145	0.135

## Data Availability

Not applicable.

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
