# Peer review of "Direct Kinetostatic Analysis of a Gripper with Curved Flexures"

_micromachines, 2022, doi:10.3390/mi13122172_

Round 1
Reviewer 1 Report
This paper presents a stiffness matrix method to solve the kinetostatic problem of planar grippers with curved beams. The established model is interesting. The following revises are required: 1. According to table I, the size of gripper is in meter scale, which is much larger the dimension of MEMS gripper. It should be explained. 2. The detailed simulation settings of ADMAS should be given, because it seems to be a rigid-flexible coupling simulation. 3. Are the linkages and flexures the same materials? Only the deformation of flexures has been presented in Fig.6 and 7.Author Response
Reviewer 1
This paper presents a stiffness matrix method to solve the kinetostatic problem of planar grippers with curved beams. The established model is interesting. The following revises are required:
According to table I, the size of gripper is in meter scale, which is much larger the dimension of MEMS gripper. It should be explained.
Thanks for the comment. The units have been scaled to MEMS range.
We verified that the results remain invariant if all lengths are divided by scale = 1E3 (from meter to millimeter). Scaling the lengths implies that the actuation forces must be divided by scale^2 = 1E6 (from Newton to micronewton). All results and figures have been checked and updated with the new values. This law of invariance can be employed to resize the mechanism to different scales.
The detailed simulation settings of ADMAS should be given, because it seems to be a rigid-flexible coupling simulation.
Thanks for the comment. The system is a rigid-flexible assembly composed of rigid links and flexible curved beams. This concept has been recalled many times in the text (for example, refer to lines 126, 135, 212, 236, Figure 4). Adams model has been developed following the same rigid-flexible coupling as the Matlab model. The flexures have been attached with fixed connections to the endpoints of the rigid links. Further details have been inserted in the text: “The two models have been compared to a model implemented using the commercial multibody software MSC Adams$^\copyright$. The model includes rigid links and flexible circular beams. The latter are combined to the links' endpoints through fixed connections. The model has been developed to be comparable with the Matlab model. The only difference pertains to the flexures since the curved beams have been modeled using a two-dimensional geometrical non-linear representation for beam-like structures.”
Are the linkages and flexures the same materials? Only the deformation of flexures has been presented in Fig.6 and 7.
The reviewer's observation is valid. As before, the deformations pertain only to the curved beams, while the remaining bodies are rigid. Our rigid link-flexible flexures assumption has been verified using FE models. It is fully justified in grippers with CSFH joints as links are at least one order of magnitude stiffer than circular beam flexures. We included the text: “Considering grippers with CSFH joints, one link is a part of the monolithic structure enclosed between two circular beams. In a MEMS device, the entire monolithic structure can deform. However, the assumption of rigid links coupled to flexible circular beams has been verified using FE models. It is fully justified since the links are at least one order of magnitude stiffer than the circular beam flexures.”

Reviewer 2 Report
Check that the style of writing is in the third person throughout. Avoid the use of ‘we’.
Though the MEMS acronym is well known to everyone as micro-electro-mechanical-systems, it should be defined when it appears first.
It is good to include a flowchart or pseudo code of the Newton-Raphson iterative method in the annexure.
What’s the drawback of the current method?
Justify the selection of a curved beam.
Comment on the sustainability of the proposed method.
The conclusion must be revised to make it clearer in terms of achievements.
The manuscript is more like a report than a research paper failing in solid discussion. Revise results and discussion part by critically examining results and including inferences drawn.
Author Response
Reviewer 2
- Check that the style of writing is in the third person throughout. Avoid the use of ‘we’.
Thanks for the comment, the text has been changed in the third person.
- Though the MEMS acronym is well known to everyone as micro-electro-mechanical-systems, it should be defined when it appears first.
Thanks for the comment, we modified the Abstract accordingly (line-1).
- It is good to include a flowchart or pseudo code of the Newton-Raphson iterative method in the annexure.
Thanks for the comment. We included a pseudo-code of the Newton-Raphson iterative method inside the “Algorithm 1” at the end of Subsection 3.2 (lines 222-223).
- What’s the drawback of the current method?
Thanks for the question. At present, the method is valid only for planar mechanisms and small deformations of the flexures.
- Justify the selection of a curved beam.
Thanks for the comment. We added the text (lines 98-105): “ Flexures employed in this context are curved beams. It has been demonstrated that curved beams can provide large rotations while maintaining small errors in terms of displacements of its center \cite{verotti2015mems}, as it is typical for classic revolute pairs. This feature is important to guarantee finite rotations of the end-effector in monolithic structures such as MEMS-based grippers. Furthermore, a linear model is capable of faithfully reproducing the displacements and in-plane rotation of the curved beam tip up to rotations of approximately $\pm20^\circ$. This feature has the considerable advantage of using a constant stiffness matrix, as will be recalled below.”
- Comment on the sustainability of the proposed method.
Thanks for the comment. The proposed theoretical method is based on matrix calculus and therefore is very suitable for implementation in programming languages. In this investigation, the developed Matlab script was found to be more versatile than the model, with similar characteristics, developed on a commercially available package of finite element and multibody dynamics. Furthermore, the proposed method can be extended to other planar mechanisms with constant stiffness flexures without changing the mathematical background.
- The conclusion must be revised to make it clearer in terms of achievements.
Thanks for the comment, conclusions and discussion have modified and combined into a single section. The following achievements have been highlighted:
- The results of Fig.13 suggest a possible extension to real-time applications of micro and nano-grippers. It is known that the control often requires simplified models to be executed quickly by the control unit. Often these models are obtained by linearizing the equilibrium equations around one or more operating points. Using models with reduced complexity would allow more efficient control strategies such as control in the operating space, inverse dynamics control, or pre-calculated torque control.
- The tangent stiffness matrix is obtained in closed form. This feature prevents the use of numeric differentiation, making the convergence process of the direct kinetostatic solution more robust. Furthermore, splitting the expression of $\Km_T$ allowed for identifying its most basic terms and calibrating the compromise between the number of iterations and calculation time. The calculation times are considerably reduced by using only the first term of the tangent stiffness matrix and recalculating it at each iteration of the Newton-Raphson algorithm described in Algorithm 1.
- The tangent stiffness matrix can be employed to develop a dynamics model o to study vibrations. The tangent stiffness matrix is the core of implicit time integration methods primarily employed in flexible multibody dynamics \cite{bauchau2011flexible}.
- Shape optimization takes further advantage of the closed form of $\Km_T$ opening scenarios to gradient-based constrained optimization problems based on the kinetostatic analysis.
- Both the two simplified models employ curved beams modeled by a constant stiffness matrix. Despite this, the curved beams guarantee finite displacements/rotations in the mechanism, allowing for the expansion of the reachable workspace. The model remains reliable for most of the mechanism's workspace. The results are accurate in the functional area except for the limit zones of the workspace in which physical constraints usually prevent motion. When the maximum rotations of the curved beams exceed $\pm20^\circ$ the constant stiffness hypothesis can no longer capture the geometric nonlinearities, and the results deviate from the actual case.
- Although the proposed method is valid only for planar cases; it can be extended to other compliant mechanisms with constant stiffness flexures without changing the mathematical background.
- The manuscript is more like a report than a research paper failing in solid discussion. Revise results and discussion part by critically examining results and including inferences drawn.
Thanks. As for the previous comment, we modified the discussion to examine the results and draw conclusions.

Reviewer 3 Report
The purpose of this paper is to solve the problem of the direct kinetostatic analysis of planar grippers with curved beams with reliable in terms of motion accuracy and actuation forces and computationally efficient to extend the formulation for real-time applications. Overall, this is a well written paper. Following issues should be addressed before recommending for publication.
1, For the Figure 1, the initial and deformed configuration should be marked carefully, which one refers initial state and which one refers deformed state, solid line or dash line?
2, The proposed method is to analysis gripper with curved flexures. However, the reviewer didn’t find any gripper models in this paper. Did this method applicable for both planar and spatial grippers?
3, How to verify the correctness of the proposed theoretical model?
4, The introduction part is not comprehensive, compliant mechanism also can be potentially applied to precision engineering like polishing or deburring.
5, The reviewer suggests citing some recently (in two years) published related papers to describe the novelty of this research.
6, The authors cite too many papers from Chen, G. It may give the readers impression that the purpose of this work is to increase the number of citations to a given author’s work.
7, Since the Pseudo-Rigid-Body Model is already shortening as PRBM in line 23, the author can write PRBM directly in line 54.
8, In Section 3, second paragraph, there is typo-Fig.??. Please have a check.
Author Response
Reviewer 3
The purpose of this paper is to solve the problem of the direct kinetostatic analysis of planar grippers with curved beams with reliable in terms of motion accuracy and actuation forces and computationally efficient to extend the formulation for real-time applications. Overall, this is a well written paper. Following issues should be addressed before recommending for publication.
- For the Figure 1, the initial and deformed configuration should be marked carefully, which one refers initial state and which one refers deformed state, solid line or dash line?
Thanks for the comment. The caption has been modified to distinguish the two configurations. “Curved beam in its initial (dashed line) and deformed configuration (solid line)”.
- The proposed method is to analysis gripper with curved flexures. However, the reviewer didn’t find any gripper models in this paper. Did this method applicable for both planar and spatial grippers?
Thanks for the comment. Most of microgrippers consist of compliant mechanisms that are fabricated by means of MEMS Technology. Many of them use lumped compliance and therefore include two or more flexure hinges. The example reported in the present investigation refers to a large class of four-bar linkage type microgripper, as the one illustrated in Figure 5a. Due to the adoption of a planar fabrication process, almost all of them are plane mechanisms.
The new Figure 5 has been included to show the full gripper. For symmetry, in the case study only half a gripper is analyzed. The proposed formulation can be applied only to planar grippers. Nevertheless, the method could be applied to other planar mechanisms and the curved beams can be replaced by other flexures with constant stiffness matrices.
- How to verify the correctness of the proposed theoretical model?
The theoretical model is based on the Lagrangian description of continuum mechanics. The proposed formulation is simplified since linear flexures are employed. Nevertheless, we demonstrated that the case study provides results in good agreement with well-established software such as Adams. The goodness of the results is further confirmed by the range of use of the mechanism. In fact, the results are accurate right where you need them, that is, in the functional area inside the workspace. It should be remembered that the extreme zones of the workspace cannot be reached for various reasons related to comb-drive pull-in phenomena, allowable stress and so on.
- The introduction part is not comprehensive, compliant mechanism also can be potentially applied to precision engineering like polishing or deburring.
Thanks, we have inserted three new papers on this topic (line 29):
%----------------------------------------------
@ARTICLE{Guo2011593,
author={Guo, T. and Li, J. and Wang, H.},
title={Application of compliant mechanism on the polishing robot},
journal={Applied Mechanics and Materials},
year={2011},
volume={63-64},
pages={593-596},
doi={10.4028/www.scientific.net/AMM.63-64.593},
note={cited By 1},
url={https://www.scopus.com/inward/record.uri?eid=2-s2.0-79959705349&doi=10.4028%2fwww.scientific.net%2fAMM.63-64.593&partnerID=40&md5=61b10d888483c06b30690548cd240533},
document_type={Conference Paper},
source={Scopus},
}
%----------------------------------------------
@ARTICLE{Zhan20131587,
author={Zhan, J. and Zhang, M.},
title={Study on edge-forming mechanism for aspheric surface compliant polishing},
journal={Zhongguo Jixie Gongcheng/China Mechanical Engineering},
year={2013},
volume={24},
number={12},
pages={1587-1590+1591},
doi={10.3969/j.issn.1004-132X.2013.12.006},
note={cited By 3},
url={https://www.scopus.com/inward/record.uri?eid=2-s2.0-84880031417&doi=10.3969%2fj.issn.1004-132X.2013.12.006&partnerID=40&md5=9d43ed0f9c87434c82311d1c1af97f35},
document_type={Article},
source={Scopus},
}
% -------------------------------------------------------------------------
@ARTICLE{Zhu2021414,
author={Zhu, W. and Liu, J. and Li, H. and Gu, K.},
title={Design and analysis of a compliant polishing manipulator with tensegrity-based parallel mechanism},
journal={Australian Journal of Mechanical Engineering},
year={2021},
volume={19},
number={4},
pages={414-422},
doi={10.1080/14484846.2019.1629678},
note={cited By 0},
url={https://www.scopus.com/inward/record.uri?eid=2-s2.0-85068047715&doi=10.1080%2f14484846.2019.1629678&partnerID=40&md5=d09ae6b2678e082b534dbea05d4177e6},
document_type={Article},
source={Scopus},
}
- The reviewer suggests citing some recently (in two years) published related papers to describe the novelty of this research.
Thanks, we included the following paper along with the previous three (line 28):
@ARTICLE{Hartmann20151,
author={Hartmann, L. and Zentner, L.},
title={A compliant mechanism as rocker arm with spring capability for precision engineering applications},
journal={Mechanisms and Machine Science},
year={2015},
volume={30},
pages={1-7},
doi={10.1007/978-3-319-15862-4_1},
note={cited By 0},
url={https://www.scopus.com/inward/record.uri?eid=2-s2.0-84943614481&doi=10.1007%2f978-3-319-15862-4_1&partnerID=40&md5=205a5265d9edb47e6cf50191b8df647f},
document_type={Conference Paper},
source={Scopus},
}
- The authors cite too many papers from Chen, G. It may give the readers impression that the purpose of this work is to increase the number of citations to a given author’s work.
We guarantee that Chen’s work has been cited only because they were pertinent to the matter and these multiple citations were not intentional. We certify that we are not even acquainted with Prof. Chen and no relation has been established with him.
- Since the Pseudo-Rigid-Body Model is already shortening as PRBM in line 23, the author can write PRBM directly in line 54.
Thanks, we modified the text (line 54).
- In Section 3, second paragraph, there is typo-Fig.??. Please have a check.
Thanks, it was a typo. It refers to Fig. 2 (line 126).

Round 2
Reviewer 1 Report
The manuscript has been revised well, and I have not further comments.
Reviewer 2 Report
Authors have addressed all my comments properly. No more comments.
Reviewer 3 Report
The author well responsed the reviewer’s comments. Why the citation of references of 20, 21 22 was added. In addition, the 21^st reference is a Chinese journal. How about using “Design of a spatial constant-force end-effector for polishing/deburring operations”to instead it.